



# GOSI9: UK Global Ocean and Sea Ice configurations

Catherine Guiavarc'h[1], David Storkey[1], Adam T Blaker[2], Ed Blockley[1], Alex Megann[2], Helene Hewitt[1], Michael J Bell[1], Daley Calvert[1], Dan Copsey[1], Bablu Sinha[2], Sophia Moreton[1], Pierre Mathiot[3,1], and Bo An[4]

[1]Met Office, FitzRoy Road, Exeter EX1 3PB, UK
[2]Marine Systems Modelling, National Oceanography Centre, Southampton, SO14 3ZH, UK
[3]Univ. Grenoble Alpes/CNRS/IRD/G-INP, IGE, Grenoble, France
[4]State Key Laboratory of Numerical Modeling for Atmospheric Sciences and Geophysical Fluid Dynamics, Institute of Atmospheric Physics, Chinese Academy of Sciences, Beijing, 100029, China

**Correspondence:** Catherine Guiavarc'h (catherine.guiavarch@metoffice.gov.uk)

**Abstract.** The UK Global Ocean and Sea Ice configuration version 9 (GOSI9) is a new traceable hierarchy of three model configurations at $1°$, $\frac{1}{4}°$ and $\frac{1}{12}°$ based on version 4.0.4 of the NEMO code. GOSI9 has been developed as part of the UK's Joint Marine Modelling Programme (JMMP), a partnership between the Met Office, the National Oceanography Centre, the British Antarctic Survey, and the Centre for Polar Observation and Modelling. Following a seamless approach it will be used for a variety of applications across a wide range of spatial and temporal resolutions: short-range coupled NWP forecasts, ocean forecasts, seasonal and decadal forecasts, climate and Earth system modelling. The GOSI9 configurations are described in detail with special focus on the updates since the previous version (GO6-GSI8). Results from 30-year ocean-ice integrations forced by CORE2 fluxes are presented for the three resolutions and the impacts of the updates are assessed using the $\frac{1}{4}°$ integrations. The upgrade to NEMO 4.0.4 includes a new sea ice model SI$^3$ (Sea Ice modelling Integrated Initiative) and faster integration achieved through the use of partially implicit schemes that allow a significant increase in the length of the time step. The quality of the simulations are generally improved compared to GO6-GSI8. The temperature and salinity drifts are largely reduced thanks to the upgrade to NEMO 4.0.4 and the adoption of $4^\text{th}$ order horizontal and vertical advections helping to reduce the numerical mixing. To improve the representation of the Southern Ocean, a scale-aware form of the Gent-McWilliams parametrisation and the application of a partial slip lateral boundary condition on momentum in the Southern Ocean have been added, resulting in a stronger and more realistic Antarctic Circumpolar Current (ACC) transport and a reduction of the temperature and salinity biases along the shelf of Antarctica. In the Arctic, the representation of sea ice is improved, leading to a reduction in surface temperature and salinity biases. In particular, the excessive and unrealistic Arctic summer sea ice melt in GO6-GSI8 is significantly improved in GOSI9 and can be attributed to the change in the sea ice model and to the higher albedos which increased sea ice thickness.



## 1   Introduction

The Joint Marine Modelling Programme (JMMP), founded in 2018, is a partnership between the Met Office and UK research centres: the National Oceanography Centre, the British Antarctic Survey and the Centre for Polar Observation and Modelling. The JMMP's ambition is to provide national capability modelling infrastructure, configurations and model output to the UK
community. Adopting the seamless forecasting approach (Brown et al., 2012), the JMMP global ocean and sea ice configurations are used for a variety of applications across a wide range of spatial and temporal resolutions: short-range coupled NWP forecasts, ocean forecasts, seasonal and decadal forecasts, climate and Earth system modelling.

This paper describes the latest Global Ocean and Sea Ice configuration, GOSI9, based on the NEMO ocean modelling framework (Madec and system team, 2019). As for the previous global ocean configuration, GO6 (Storkey et al., 2018),
GOSI9 is a traceable hierarchy of three resolutions: $1°$, $\frac{1}{4}°$ and $\frac{1}{12}°$. Previously, new configurations of the Global Ocean (GO, Storkey et al., 2018; Megann et al., 2014) and Global Sea Ice (GSI, Ridley et al., 2018) were released separately, in part due to the dependence of the two models on different source codes. From NEMO version 4 (Madec and system team, 2019), the new native sea ice model SI$^3$ (Sea Ice modelling Integrated Initiative) is available as part of the NEMO ocean modelling framework, and from this release onwards will be used for JMMP configurations. In the early stages of this development
cycle the configuration was referred to as GO8p[0-7], and some papers based on these pre-release versions may have used this naming convention. However, it was realised that the previous release of the sea ice component had already reached version 8, and a second release at version 8 could lead to confusion. To explicitly recognise the joint release of both ocean and sea ice components we have adopted a new nomenclature for joint releases, GOSI.

Development of new configurations is motivated primarily by the desire to improve the representation of the ocean and sea
ice, and secondarily to deliver better (more efficient) computational performance. The latter often comes from adopting newer code, whereas the former more frequently requires a combination of expert assessment to identify the cause of biases and to identify or write improved numerical schemes and an element of tuning/calibration to optimise the solution. Warm SST biases in the Southern Ocean have been a long standing issue in the Met Office coupled systems. In the Met Office GC3 coupled configuration (HadGEM3) based on the GO6+GSI8 ocean and sea ice components, the Southern Ocean SST bias was reduced
compared with GC2 thanks to improved representation of clouds Williams et al. (2018) and tuning of the isopycnal diffusion parameter Storkey et al. (2018). However a warm bias still remained (Roberts et al., 2019). Hewitt et al. (2016) and Roberts et al. (2019) looked at the representation of the Southern Ocean in the hierarchy of HadGEM3 coupled models. They highlight the large biases in the Antarctic Circumpolar Current (ACC) transport especially in $\frac{1}{4}°$ configuration which has the lowest transport, reduced by 40% compared to observations.

GOSI9 is required to deliver improved performance in both forced (ocean and sea ice only) and coupled configurations. It will form the ocean and sea ice component of the forthcoming GC5 version of the Met Office Hadley centre coupled climate model (Xavier et al., 2023) and will be the physical basis of the next UKESM (Mulcahy et al., 2023). To mitigate the risk of having to introduce configuration changes late in the development cycle a new approach was adopted: GOSI9 $\frac{1}{4}°$ was tested in a coupled configuration at an early stage during the development process. This revealed some key insights into potential mean



state biases that we were then able to address during the cycle. We describe this process later, but the main focus of this paper is
    to present the results and testing of GOSI9 forced by atmospheric surface forcing. Evaluation of the final coupled configuration
    is presented in (Xavier et al., 2023).

    The paper is organised as follows. Section 2 provides a detailed description of the GOSI9 configurations and the develop-
    ments introduced into the global ocean configurations since the previous release, GO6. Section 3 describes the model simula-
tions presented in this study, including the forcing and initialisation used. In Section 4 we present the results from the global
    evaluation for the three resolutions comparing GOSI9 and GO6, and the impact of individual changes are assessed using the
    $\frac{1}{4}^{\circ}$ degree configuration. Section 5 provides a more detailed evaluation of three key regions of interest: North Atlantic, North
    West Pacific and Southern Ocean.

## 2   Model description

GOSI9 is based on NEMO v4.0.4 (Madec and system team, 2019) and the SI$^3$ sea ice component (Blockley et al., 2023) and
    is built on the traceable hierarchy of three model configurations at $1^{\circ}$, $\frac{1}{4}^{\circ}$ and $\frac{1}{12}^{\circ}$ horizontal resolutions described in Storkey
    et al. (2018). For ease of reference, in this section we provide a complete description of GOSI9. Details of the configuration that
    remain common to GO6, namely the grids, bathymetries, free surface solution and advection, mixing and boundary conditions
    are described first. We then describe the developments that have been made since GO6. Together these define the GOSI9
configuration.

### 2.1   Model grids and bathymetries

    The model grids and bathymetries are unchanged from GO6 (Storkey et al., 2018). For the three resolutions, the grids are
    based on the ORCA global grids available on the NEMO framework (Madec and system team, 2019). eORCA1, eORCA025
    and eORCA12 have a nominal $1^{\circ}$, $\frac{1}{4}^{\circ}$ and $\frac{1}{12}^{\circ}$ resolution at the Equator and an isotropic Mercator grid (i.e. same zonal
and meridional grid spacing).To avoid singularity point in the ocean, in the Northern Hemisphere, the grids are quasi-isotropic
    bipolar with two north mesh poles being introduced on lands in Siberia and Canada. To better represent the equatorial dynamics,
    from $20^{\circ}$ N/S the eORCA1 model grid meridional resolution increases towards $\frac{1}{3}^{\circ}$ at the Equator. In the Southern Hemisphere,
    south of $67^{\circ}$ S, the grid is extended to $85^{\circ}$ S following the method described in Mathiot et al. (2017) to include ice shelf
    cavities.

The three models have the same vertical grid with 75 vertical levels. The level thickness is double tanh function of depth
    increasing from 1m thickness near the surface to 200m at 6000m depth. It provides a high resolution near the surface to resolve
    ocean responses to atmospheric forcing, including the diurnal cycle (Bernie et al., 2005) and a reasonable resolution at mid-
    depths for long term climate responses. Stewart et al. (2017) show that 75 levels is the minimum number capable of resolving
    the second baroclinic mode and the surface layer. Partial step topography is used, making the depth of the bottom cell variable
and adjustable to the real depth of the ocean (Adcroft et al., 1997; Barnier et al., 2006).





Different data sets were used to produce the model bathymetries for each resolution. In this regard, the hierarchy of resolutions is not fully traceable. For the 1° resolution model, eORCA1, the bathymetry is derived from ETOPO2 data set (National Geophysical Data Center, 2006) with additional data from IBSCO (Arndt et al., 2013) on the Antarctic shelf. For the $\frac{1}{4}°$ resolution model, eORCA025, the bathymetry is derived from the ETOPO1 data set (Amante and Eakins, 2009) with additional data from GEBCO (IOC et al., 2003) in coastal regions and from IBSCO (Arndt et al., 2013) on the Antarctic shelf. For the $\frac{1}{12}°$ resolution model, eORCA12, the bathymetry is derived from GEBCO_2014 (Weatherall et al., 2015).The bathymetry for eORCA025 had gridscale smoothing applied. This was not done for the eORCA1 or eORCA12 bathymetries. A traceable set of bathymetries derived from a common source is planned for the next development cycle.

### 2.2 Free surface solution and advection

As in GO6, in order to represent accurately the surface fresh water flux, the model uses a nonlinear free surface allowing cell thicknesses throughout the water column to vary with time (z* coordinate as in Adcroft and Campin, 2004). A change from GO6 (Storkey et al., 2018), in which a time-filtering solution was used with the fastest waves being filtered (Roullet and Madec, 2000), is that the equation of the surface pressure gradient is solved using the split-explicit free surface formulation, also called time-splitting formulation, following Shchepetkin and McWilliams (2005). The time-splitting solution allows for an explicit representation of the fastest external gravity waves.

For momentum advection, GOSI9 uses the vector-invariant form in which the horizontal advection is separated into rotational and irrotational terms. The vorticity term (including the Coriolis term) is calculated using the energy and enstrophy conserving scheme of Arakawa and Lamb (1981). In NEMO, different options of this scheme are available differing in the way the topographic boundary condition is represented. We retain the option used in GO6, nn_een_e3f=0, which tends to reinforce the topostrophy of the flow (Madec and system team, 2019, Sect. 5.2.1). The irrotational part of the momentum advection is formulated according to Hollingsworth et al. (1983) in order to avoid near grid-scale horizontal numerical instabilities (Ducousso et al., 2017) . Advection of tracers is performed using the total variance diminishing (TVD) scheme of Zalesak (1979) with $4^{th}$ order on horizontal and vertical directions. In GO6, the same scheme was used but with $2^{nd}$ order on horizontal and vertical directions.

### 2.3 Mixing and boundary conditions

Lateral diffusion of momentum is on geopotential surfaces and, as in GO6, uses a Laplacian viscosity in eORCA1 and a bi-Laplacian viscosity in eORCA025 and eORCA12. The coefficients are specified in Table 1. In the bi-Laplacian case, to prevent instabilities due to numerical diffusion, the viscosity coefficients reduce polewards with the cube of the grid length. For the eORCA1 model, the viscosity coefficients reduce linearly with the increased meridional grid spacing towards the Equator, but are constant poleward of 20° N/S. This was an error in the input file which remains unchanged from GO6, and was identified too late in the development cycle to redress. Grid space dependent values have been tested and found to give small improvements in model fidelity. A modification to the input file similar to that applied by Hutchinson et al. (2023) will be necessary for applications in which the Antarctic ice shelf cavities are opened.





Lateral diffusion of tracers is performed along isoneutral surfaces using Laplacian mixing with coefficients given in Table
1. A parameterisation of adiabatic eddy mixing (Gent and Mcwilliams, 1990) with a spatially varying coefficient (Held and
Larichev, 1996; Tréguier et al., 1997) is used in eORCA1. A weak grid-scale aware Gent and Mcwilliams (1990) parameteri-
sation is now applied at the higher resolutions (see section 2.5.6).

The mixing parameterisations have been mainly unchanged compared to GO6. The vertical mixing of tracers and momentum
is parameterized using a modified version of the Gaspar et al. (1990) turbulent kinetic energy (TKE) scheme (Madec and
system team, 2019). To represent unresolved mixing due to internal wave breaking, a background vertical eddy diffusivity and
a background viscosity are both applied. The background vertical eddy diffusivity has a value of $1.2 \times 10^{-5}$ m$^2$ s$^{-1}$, which
decreases linearly from $\pm 15°$ latitude to a value of $1.2 \times 10^{-6}$ m$^2$ s$^{-1}$ at $\pm 5°$ latitude (Gregg et al., 2003). The background
viscosity is applied globally with a constant value of $1.2 \times 10^{-4}$ m$^2$ s$^{-1}$. Other unresolved processes are represented using
parametrisations available in NEMO. Surface wave breaking mixing is parametrised following Craig and Banner (1994),
increasing the mixing at the surface. Axell (2002) is used to represent the Langmuir cell mixing. An ad-hoc parametrisation
is used to represent the mixing due to near-inertial wave breaking (Rodgers et al., 2014) with a length scale which can be
varied geographically. Extensive work was carried to tune this length scale in GO6 (Storkey et al., 2018). Additional tuning
was carried out during the development of GOSI9 in order to reduce biases in the coupled configuration (see Section 2.5.8).

To parametrise the convection, an increased vertical diffusivity of 10 m$^2$ s$^{-1}$ is applied where the water column becomes
unstable. The double diffusive mixing is parametrised using Merryfield et al. (1999). A climatological geothermal heat flux
(Stein and Stein, 1992) is added as a bottom boundary condition. A quadratic bottom friction is used globally with enhanced
coefficient in the Indonesian Throughflow, Denmark Strait and Bab-el-Mandeb regions. Following Beckmann and Döscher
(1997), an advective and diffusive bottom boundary layer scheme is used. The tidal mixing is parameterised following Simmons
et al. (2004) with a special formulation for the Indonesian Throughflow as recommended by Koch-Larrouy et al. (2008).

**2.4 Fresh water input from land**

Fresh water flux from the river runoffs is applied to the ocean surface layer. The freshwater runoffs are considered as fresh and as
having the same temperature as the local SST. In order to avoid instabilities caused by shallow fresh layers, the vertical mixing
is increased to $2 \times 10$ m$^2$ s$^{-1}$ in the top 10 m of the water column at runoff locations. Along Antarctica, to represent the icesheet
fresh water input, we apply a parameterisation of ice shelf basal melting. Following Mathiot et al. (2017), climatological
fresh water input is prescribed at the edge of the ice shelves through depth, to mirror the effect of ice shelf basal melt on the
circulation. Due to stability issues in GOSI9 we use iceberg melt climatology based on GO6 integrations instead of interactive
icebergs (see section 2.5.7).





|  | eORCA1 | eORCA025 | eORCA12 |
|---|---|---|---|
| Lateral diffusion of momentum | laplacian | bilaplacian | bilaplacian |
| Lateral viscosity | From 3D file, 20,000 $m^2$ $s^{-1}$ poleward of $20°$ N/S, reducing with meridional grid size towards the Equator | Eddy coefficient varying with grid size With Lateral viscous velocity (rn_uv) 0.0838 | Eddy coefficient varying with grid size With Lateral viscous velocity (rn_uv) 0.1895 |
| Lateral diffusive velocity (rn_ud) | 0.018 | 0.011 | 0.027 |

**Table 1.** Parameter changes between eORCA1, eORCA025 and eORCA12 configurations.

## 2.5 Development and changes since GO6

### 2.5.1 Upgrade to NEMO 4.0.4

For GOSI9, the version of the NEMO code has been upgraded to NEMO 4.0.4 (compared to NEMO 3.6 for GO6). NEMO 4.0 was released in 2019 (Madec and system team, 2019). GOSI9 uses a split-explicit free surface proposed by Shchepetkin and McWilliams (2005). With the time split solution, external gravity waves are explicitly represented. GO6 used the filtered free surface available at NEMO v3.6 (Roullet and Madec, 2000) where the fastest waves are filtered. With the time split solution the fast barotropic motions (such as tides) are also simulated with a better accuracy. In NEMO4.0, the lateral diffusion code

has been rewritten with a different formulation including scale aware setting of eddy viscosity and diffusivity.

The AeroBULK package (Brodeau et al., 2017) has been implemented in NEMO 4.0. With AeroBULK, four bulk formulations are available in NEMO. GOSI9 uses the NCAR formulation (formerly CORE) appropriate for forcing the model with the CORE2 dataset, the same formulation used in GO6 but AeroBULK provides some refinements with the computation of the air density and with the reduction of approximation in the estimation of surface specific humidity of saturation by adding a

dependence to the sea level pressure.

### 2.5.2 Sea Ice model SI[3]

The sea ice component of GOSI9 is based upon NEMO's new native sea ice model, SI[3] (Sea Ice modelling Integrated Initiative). SI[3] is a dynamic-thermodynamic continuum sea ice model that includes an ice thickness distribution (ITD), conservation of horizontal momentum, an elastic-viscous plastic (EVP) rheology, and energy-conserving halo-thermodynamics (Vancop-

penolle et al., 2023). SI[3] has been available in the NEMO code since version 4.0, having been created by merging functionality from several different sea ice models used with NEMO (namely LIM, CICE, and GELATO), building upon the LIM3 model of Rousset et al. (2015). SI[3] is fully embedded within NEMO and is invoked from within NEMO's Surface Boundary Code (SBC) module.




Aside from the change in model, the physics of the sea ice component of GOSI9 are largely similar as used in the previous
GO6+GSI8 configuration (Ridley et al., 2018; Storkey et al., 2018) based upon CICE5. GOSI9 uses 5 thickness categories to
model the sub-grid ITD, with an additional ice-free category for open water (Thorndike et al., 1975). Thermodynamic growth
and melt of the sea ice is modelled using multi-layer thermodynamics, with 4 layers of ice and one of snow based upon Bitz and
Lipscomb (1999). Sea ice dynamics are modelled using the elastic–viscous–plastic (EVP) rheology of Hunke and Dukowicz
(2002). The biggest difference compared with GO6+GSI8 (Storkey et al., 2018) is the use of the broadband albedo scheme in
SI3 instead of the dual-band scheme used in CICE. This has allowed us to tune the albedo independently of the coupled model
and we increased the values of the albedo relative to the SI3 defaults (ponded ice albedo is increased from 0.27 to 0.36, dry ice
albedo is increased from 0.60 to 0.70 and dry snow albedo is increased from 0.85 to 0.87). The sea ice component of GOSI9 is
run on the same model grid as the ocean and at every ocean time step. More details on the specifics of the sea ice configuration,
along with details of how SI$^3$ was adapted to work in the HadGEM3 coupled model, can be found in Blockley et al. (2023).

### 2.5.3 Equation of state TEOS-10

In GOSI9, the equation of state was upgraded to the Thermodynamic Equation Of Seawater – 2010 (TEOS-10, Ioc et al.,
2010) instead of the previous standard, the 1980 equation of state (EOS-80). An important change is the use of absolute
salinity and conservative temperature instead of practical salinity and potential temperature for EOS-80. TEOS-10 provides
a complete thermodynamically consistent representation of all thermodynamic properties of seawater and allows for a more
accurate representation of the heat content.

However, the use of TEOS-10 has an impact for users. For the FOAM ocean forecasting system, the observations assimilated
in the system are EOS-80 variables meaning that the model TEOS-10 variables need to be converted to EOS-80 prior to being
passed to the observation operator. There is no practical impact on coupling with atmosphere and sea ice, as when using
NEMO with TEOS-10 equation of state, the sea surface conservative (TEOS-10) temperature is converted to potential (EOS-
80) temperature before coupling.

Note that for this paper, the conservative temperature and absolute salinity fields from GOSI9 have been converted to po-
tential temperature and practical salinity to facilitate the comparison with GO6. The temperature and salinity results presented
throughout this paper are the EOS-80 variables (potential temperature and practical salinity).

### 2.5.4 Time step and performance

A benefit from upgrading to NEMO 4 has been the implementation of the adaptive-implicit vertical advection (Shchepetkin,
2015). As for most ocean models, the time step in NEMO needs to satisfy multiple criteria to maintain numerical stability.
The vertical CFL criterion is commonly the most limiting and imposes time and space discretisation constraints. Treating
the vertical advection implicitly can reduce these restrictions but causes large dispersive errors. With adaptive-implicit vertical
advection, the implicit scheme is only used in targeted areas where potential breaches of vertical CFL condition occur. It allows
a much longer time step while retaining the accuracy of the explicit scheme.



| Time step (min) | eORCA1 | eORCA025 | eORCA12 |
|---|---|---|---|
| GO6 | 45 | 20 | 6 |
| GOSI9 | 60 | 30 | 10 |

**Table 2.** Time step in minutes for all GO6 and GOSI9 configurations.

Adaptive-implicit vertical advection has been introduced in the GOSI9 configurations as well as an implicit sea ice drag (available from NEMO 4.0.4). These changes allow the use of a considerably longer time step (Table 2) without introducing any significant changes or biases in the ocean and sea ice (not shown). For the $\frac{1}{4}^{\circ}$ model eORCA025, the time step has been increased by 50% in both ocean-only and coupled (GC5) configurations. For the $\frac{1}{12}^{\circ}$ configuration eORCA12, the time step has been increased by 66%, allowing to produce 2 years of simulation per day on 6150 cores. For the 1° configuration eORCA1, the time step has been increased by 33%.

### 2.5.5 Numerical mixing

Numerical mixing, caused mainly by truncations in the tracer advection scheme, is a recognised problem in ocean models. In models such as NEMO that use quasi-Eulerian vertical coordinates it arises from errors in vertical advection associated with internal waves and tides, and it also results from horizontal tracer advection in regions with a high cell Reynolds number where the mesoscale is not well resolved, which are particularly extensive on a $\frac{1}{4}^{\circ}$ grid at mid and high latitudes. Megann (2018) estimated the numerical mixing in a $\frac{1}{4}^{\circ}$ GO5.0 NEMO configuration by evaluating an effective diffusivity from the density transformation rate, and demonstrated that the effective diffusivity was over five times as large as the explicit diffusivity calculated in the model mixing scheme. Megann and Storkey (2021) found that increasing the viscosity in a $\frac{1}{4}^{\circ}$ GO6, either by using larger values for the fixed biharmonic viscosity parameter or changing to the Smagorinsky viscosity formulation, led to a reduction by between 10 and 20% in the effective diffusivity over much of the ocean interior, along with comparable reductions in temperature and salinity biases. The sole disadvantage of increased viscosity is a reduction in stability: in GO6, tripling the fixed viscosity required the time step to be reduced by 50% from the default of 1,350 s, while GOSI9 requires a 50% reduction in time step from the default of 1,800 s if either the fixed viscosity is doubled or the Smagorinsky scheme is selected. This increase in running cost ruled out the use of viscosity as a tool to reduce numerical mixing in coupled applications of this ocean configuration. The z time-filtered Arbitrary Lagrangian-Eulerian coordinate (Leclair and Madec, 2011) has been shown to significantly reduce numerical mixing from internal waves and tides (Megann et al., 2022; Megann, 2024), and may be considered for inclusion in future global Global Ocean and Sea Ice configurations, although this again requires a reduced timestep at present for stability.

As an alternative approach to reducing the numerical mixing resulting from advection in poorly-resolved mesoscale flows, the tracer advection in both horizontal and vertical directions was changed from second order (the default in GO6) to fourth order. This incurred no significant penalties in run time, and did not require any reduction in time step.



### 2.5.6 Southern Ocean tuning

As highlighted in the Introduction, the HadGEM3 coupled models with eddy-permitting ($\frac{1}{4}^{\circ}$) or eddy-resolving ($\frac{1}{12}^{\circ}$) ocean
resolution have a history of large-scale biases in the Southern Ocean. The largest biases appear at eddy-permitting resolution,
which shows weak ACC transports (Hewitt et al., 2016; Roberts et al., 2019) as well as overly active subpolar gyres in the
Weddell and Ross Seas and biases in the temperatures and salinities on the Antarctic shelves. Experiments in the coupled
model showed that these biases are linked, and that a package of changes aimed at damping the overactive gyres also had the
effect of improving the ACC transport and the shelf temperature and salinity biases. The work in the coupled model is described
in more detail in Storkey et al. (2024); in this paper (Section 5.1) we show the impact in the forced GOSI9 configurations.

The so-called Southern Ocean package of changes to the eddying models consists of the introduction of a scale-aware form
of the Gent-McWilliams parametrisation and the application of a partial slip lateral boundary condition on momentum in the
Southern Ocean.

Hallberg (2013) discusses ocean models that resolve eddies in parts of the domain but not in others. This is the case for
the $\frac{1}{4}^{\circ}$ and $\frac{1}{12}^{\circ}$ models presented here, in which eddies are well resolved at low latitudes but not at high latitudes due to the
decrease of the Rossby radius of deformation with latitude (see Hallberg (2013) Figure 1). At GOSI9 we try to account for
the effect of the unresolved eddies at high latitudes by introducing the same space- and time-dependent version of the Gent-
McWilliams scheme (Tréguier et al., 1997) as used in the low resolution model but with a coefficient that is capped to be zero
at low latitudes, ramping up to a low value of 75 m$^2$ s$^{-1}$ at high latitudes.

As with GO6, a free slip lateral boundary condition on momentum is applied at all resolutions. To further damp the Southern
Ocean gyres in the $\frac{1}{4}^{\circ}$ and $\frac{1}{12}^{\circ}$ configurations, we increase the topographic drag by introducing partial slip boundary condition
partial-slip condition (see Madec and system team (2019) section 7.1) south of 50°S.

### 2.5.7 Iceberg climatology

GO6 (Storkey et al., 2018) uses a Lagrangian iceberg model (Bigg et al., 1997; Martin and Adcroft, 2010). The same Lagrangian
iceberg model was deployed in GOSI9 but interactive icebergs caused stability problems. An initial stability issue caused by
excess melting was resolved by changing the melting temperature of the iceberg to the freezing point.

Stability was especially an issue with the coupled configuration, GC5, where to conserve fresh water, excess precipitation
over Antarctica is balanced through the iceberg calving. At times this resulted in double the number of icebergs compared with
the forced GOSI9, and accumulation of icebergs along the Antarctic peninsula caused regular crashes. Work was carried out
to improve realism and stability: implementation of calving distribution along the ice shelf rather than on a single point; and
implementation of a speed limiter for icebergs to prevent icebergs from travelling more than half a grid cell into the processor
halo region in one time step. These changes reduced the frequency of crashes but not to a satisfactory level. It is hoped that
further development can be done to improve stability.

In the meantime a fresh water iceberg climatology is used instead of the Lagrangian iceberg model. Tests were carried out
using the Antarctic icebergs melt climatology from Merino et al. (2016). However, Merino et al. (2016)'s climatology does



not include melt contributions from the Northern Hemisphere. Therefore we built a monthly fresh water iceberg climatology using the iceberg fresh water outputs from 30 year integrations of GO6 (Storkey et al., 2018), which are initialised from EN4 climatology and start from a state of rest with no icebergs. It takes around 4 years for the number of icebergs and the iceberg melt to stabilise, so the first five years of the simulations were discarded to create the climatology. A fresh water iceberg melt

climatology was created for each resolution: $1°, \frac{1}{4}°$ and $\frac{1}{12}°$ from the corresponding GO6 runs. The heat content of the melt water into the ocean is calculated using the freezing point temperature and the latent heat of melting is extracted from the ocean.

Figure ?? shows the annual mean fresh water flux from icebergs calculated from GO6 integrations. The distribution of iceberg melt is similar between the different resolutions. As the distribution depends closely on the circulation, the distribution

for the higher resolution configuration features more small scales with iceberg melt occurring closer to the coast. The eORCA1 iceberg melt climatology differs from the higher resolution in the Weddell Sea which could be a consequence of the difference in circulation in that region.

### 2.5.8   Coupled model GC5 tuning

During the early testing phase with the coupled model GC5, biases arose in the Indian Ocean, with too cold SST and too warm

subsurface temperatures. To better understand and to try to reduce these biases several sensitivity experiments were carried out, and these resulted in two key changes: an increase to the globally uniform value of the chlorophyll concentration; and a reduction in the TKE mixing depth between 10° S and 40° S. Increasing the chlorophyll concentration acts to reduce the depth of the solar penetrating radiation, warming the surface layer and cooling the subsurface. The chlorophyll concentration was increased from 0.05 mg.m$^{-3}$ at GO6 to 0.1 mg.m$^{-3}$, which better matches the observed climatological value in the tropics. As

part of the GO6 development, extensive work was carried to tune the near surface mixing (Storkey et al., 2018). Tuning of the e-folding length scale nn_htau associated with the parameterisation of near-inertial wave breaking resulted in the choice of a larger length scale in the Southern Ocean (Storkey et al., 2018, Fig. 2) where summertime climatological (de Boyer Montégut et al., 2004) mixed layers are on average deeper than in the northern latitudes. Revisiting this work we found that reducing the mixing depth between 10° and 40° S improved the warm subsurface bias in the Indian Ocean. The changes were tested

in the forced $\frac{1}{4}°$ GOSI9 configuration before being implemented as standard in the three GOSI9 configurations. In future configurations, we plan to use a seasonal climatology of the chlorophyll concentration which will better represent the large variations associated with spring blooms.

### 3   Integrations

For the experiments described in this paper, the model initial conditions for temperature and salinity are from monthly clima-

tologies based on EN4 objective analysis (Good et al., 2013) for the years 1995-2014. For the 30 year integrations presented in Section 4, the sea ice is initialised with a restart produced after 1 year of integration. The 1 year integration is initialised with



| Experiment | GOSI | NEMO version | Sea Ice model | SO pack. | Eq. of State | Adv. order | Chl. conc. (mg.m$^{-3}$) | TKE mixing depth | length |
|---|---|---|---|---|---|---|---|---|---|
| GO6 | GO6 | 3.6 | CICE | no | EOS80 | 2nd order | 0.05 | Storkey et al. (2018) | 1976-2005 |
| GO6 4.0 | GO6 | 4.0 | SI$^3$ | no | EOS80 | 2nd order | 0.05 | Storkey et al. (2018) | 1976-2005 |
| GOSI9 2nd order | GOSI9 | 4.0.4 | SI$^3$ | yes | TEOS10 | 2nd order | 0.1 | sect. 2.5.8 | 1976-2005 |
| GOSI9 nn_etau | GOSI9 | 4.0.4 | SI$^3$ | yes | TEOS10 | 4th order | 0.1 | Storkey et al. (2018) | 1976-1984 |
| GOSI9 chl | GOSI9 | 4.0.4 | SI$^3$ | yes | TEOS10 | 4th order | 0.05 | sect. 2.5.8 | 1976-1994 |
| GOSI9 | GOSI9 | 4.0.4 | SI$^3$ | yes | TEOS10 | 4th order | 0.1 | sect. 2.5.8 | 1976-2005 |

**Table 3.** List of experiments with eORCA025 configuration. "SO pack." stands for Southern Ocean package including GM eddy parametrisation and increased topographic drag (see section 2.5.6). "Adv. order" stands for order of the advection used in the model. The same order is used for the vertical and the horizontal advection. "Chl. conc." stands for chlorophyll concentration. For all integrations, a constant is used.

temperature and salinity from EN4 climatology and the sea ice initial state is calculated by SI$^3$ from the initial temperature and salinity. The model is spun up from a state of rest.

GOSI9 integrations are forced over the period 1976-2005 by the CORE2 surface data set (Large and Yeager, 2009) using the NCAR bulk formulae (Large and Yeager, 2009). Relative wind stress between the wind and the ocean current is used. As temporal resolution of the CORE2 forcing is not sufficient to resolve the diurnal forcing (Bernie et al., 2007), an artificial diurnal cycle is imposed on the daily mean shortwave fluxes. A sea surface salinity (SSS) restoration towards monthly mean climatology is applied with a -33.333 mm/day/psu restoring coefficient. A monthly climatology is applied for the fresh water flux from river runoff (Bourdallé-Badie and Treguier, 2006), icebergs (see Section 2.5.7) and ice shelves (Rignot et al., 2013).

Sensitivity experiments with the $\frac{1}{4}^\circ$ configuration have been carried out to test the impact of changes detailed in Section 2.5. The list of experiments is detailed in Table 3. For all experiments, initialisation and forcing follow the protocol used for GOSI9 and described above. Results from these experiments are assessed in section 4. Mixed layer depth climatology from the de Boyer Montégut et al. (2004) dataset, sea surface temperature climatology from ESA CCI (Merchant et al., 2014), temperature and salinity climatologies from EN4 objective analyses (Good et al., 2013) are used for comparison. EN4 temperature and salinity are averaged over the period 1995-2014, covering a period where data from Argo floats are available.



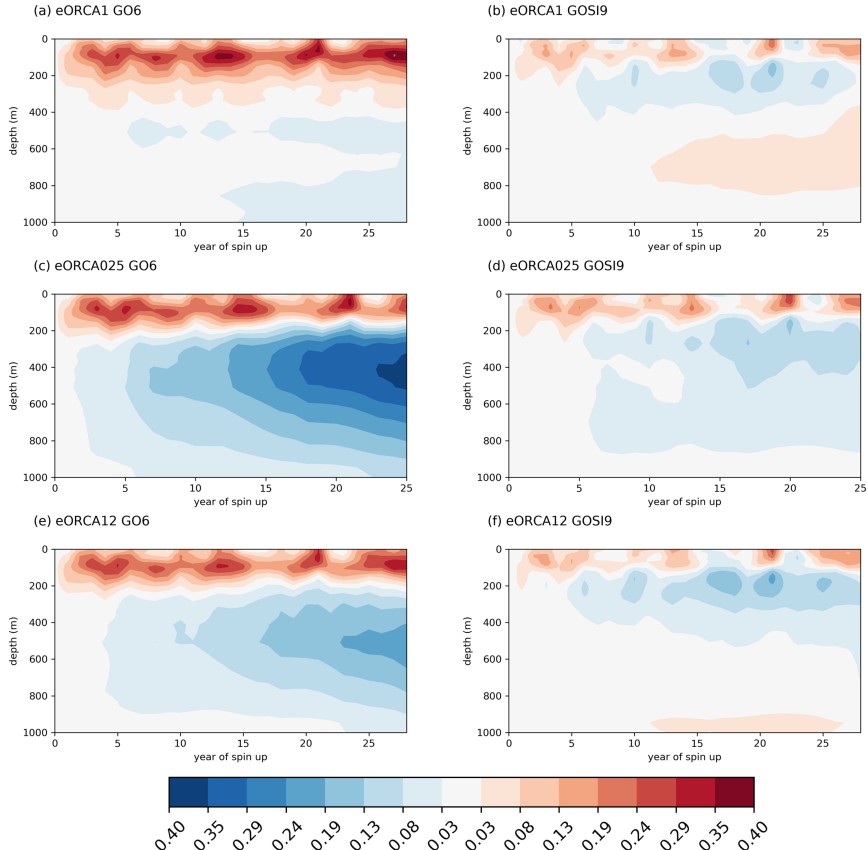

**Figure 1.** Global mean potential temperature drift from initial conditions (K) for GO6 (left) and GOSI9 (right).

## 4 Model evaluation and comparison with GO6

In this section, the results from the three GOSI9 integrations are evaluated and compared against the GO6 integrations and observation-based products.

### 4.1 Global drift and budget analysis

The first metrics used to evaluate the model performance are the drifts in the globally integrated temperature and salinity from the initial conditions (Fig. 1 and 2). Starting from an observed climatological state, the model should ideally exhibit no strong trend over time. Fig. 1 and 2 compare the drifts in GO6 (left) and GOSI9 (right) for $1°$ (a, b), $\frac{1}{4}^{\circ}$ (c, d), and $\frac{1}{12}^{\circ}$ (e, f).

GOSI9 represents a substantial improvement in both temperature and salinity biases over the top $1000\,\mathrm{m}$ compared with GO6. The warm bias which established quickly in the upper $300\,\mathrm{m}$ in GO6 is reduced to around $\frac{1}{3}$ its size across the three

resolutions, and the strong cooling that was manifest primarily in the $\frac{1}{4}^{\circ}$ and $\frac{1}{12}^{\circ}$ resolutions is far smaller. The improvements



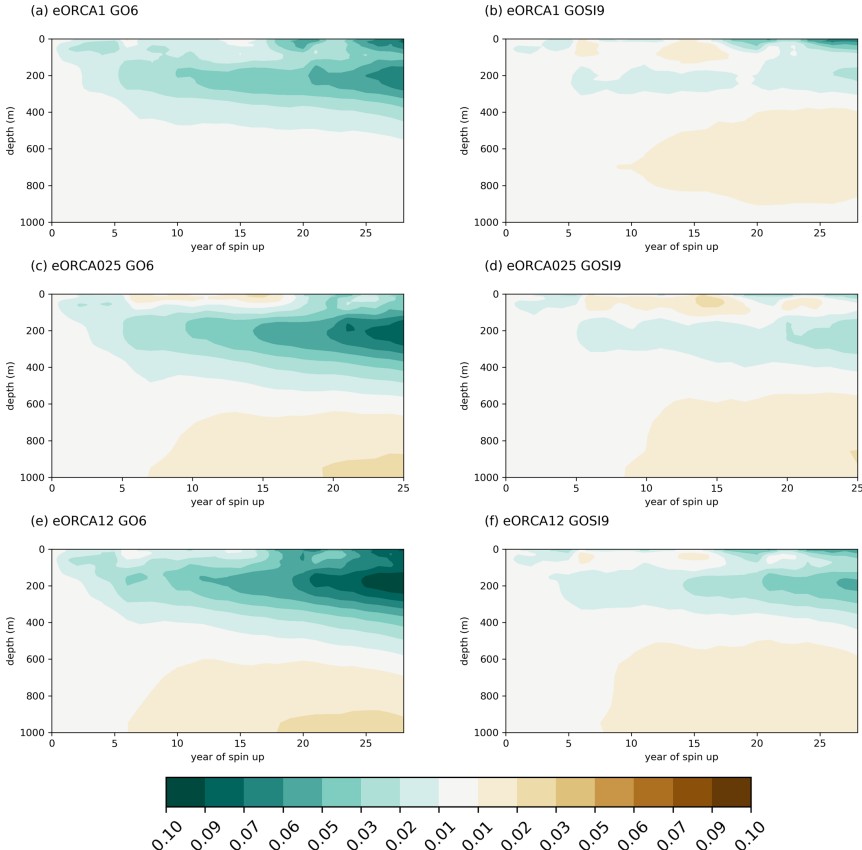

**Figure 2.** Global mean practical salinity drift from initial conditions for GO6 (left) and GOSI9 (right).

in the upper 300 m are largely attributable to the increase of chlorophyll concentration and the reduction of the mixing depth (see section 2.5.8), whilst reduction of the numerical mixing from using the 4[th] order advection scheme and the upgrade to NEMO 4.0.4 are the leading contributors to the reduction in cooling below 300 m. Figure 3 shows an evolution of the globally integrated temperature drift from GO6 (a) through to GOSI9 (f) in the $\frac{1}{4}^{\circ}$ configuration. Figures 3b-e show intermediate stages

of the development process, enabling us to attribute relative contributions to the improvement to individual changes that have been made to the model configuration. The panels correspond to the list of experiments shown in table 3.

Similarly, the fresh bias which develops over the upper 500 m, centred around 200 m depth, in GO6 is reduced by 50% or more (Fig. 2). This is mainly due to the upgrade to NEMO 4.0.4 (Fig. 4b) and the more accurate bulk formulae, which produces a salinification in the Tropics. The other developments resulted in minimal impact on the globally integrated salinity trends

(Fig. 4). While in GO6 the amplitude of the salinity and temperature drifts varies with the resolutions, all three GOSI9 models show consistent drift with little variation between resolutions.





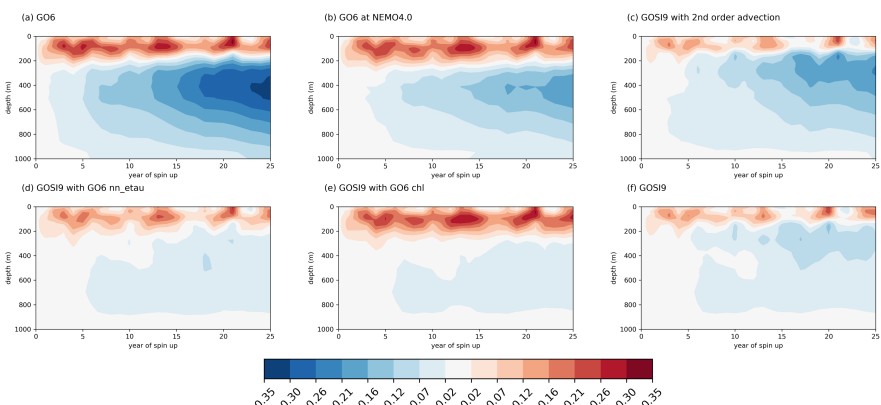

**Figure 3.** Global mean potential temperature drift from initial conditions (K) for the experiments with the $\frac{1}{4}^{\circ}$ resolution model. (a) GO6. (b) GO6 4.0: GO6 upgraded to NEMO 4.0.4, this configuration is the first experiment using the se ice model $SI^3$ and has large sea ice biases impacting the surface salinity. (c) GOSI9. (d) GOSI9 nn_etau: GOSI9 with GO6 TKE mixing depth. (e) GOSI9 chl: GOSI9 with GO6 chlorophyll concentration. (f) GOSI9 2nd order: GOSI9 with $2^n d$ order advection in horizonal and vertical. The experiments are detailed in Table 3.

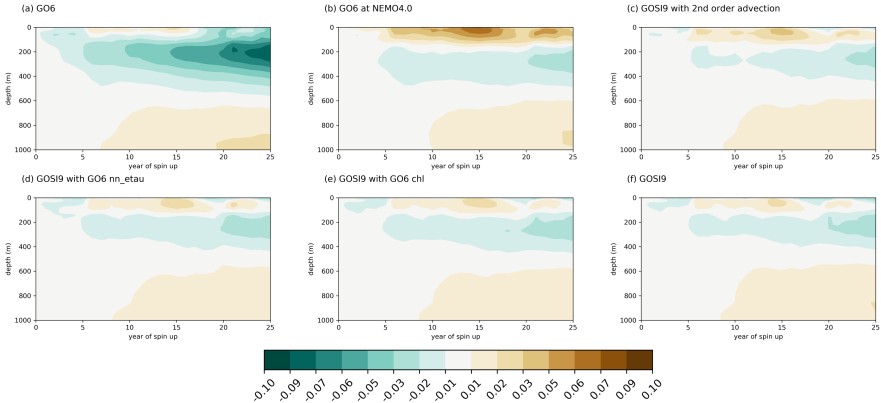

**Figure 4.** Global mean practical salinity drift from initial conditions (K) for the experiments with the $\frac{1}{4}^{\circ}$ resolution model. (a) GO6. (b) GO6 4.0: GO6 upgraded to NEMO 4.0.4, this configuration is the first experiment using the sea ice model $SI^3$ and has large sea ice biases impacting the surface salinity. (c) GOSI9. (d) GOSI9 nn_etau: GOSI9 with GO6 TKE mixing depth. (e) GOSI9 chl: GOSI9 with GO6 chlorophyll concentration. (f) GOSI9 2nd order: GOSI9 with $2^n d$ order advection in horizonal and vertical. The experiments are detailed in Table 3.

This assessment of model drift in globally integrated quantities is very positive, suggesting significant improvements over the previous generation of the models. However, we now need to assess whether these improvements arise from a reduction in biases or whether they are the product of large competing regional biases.





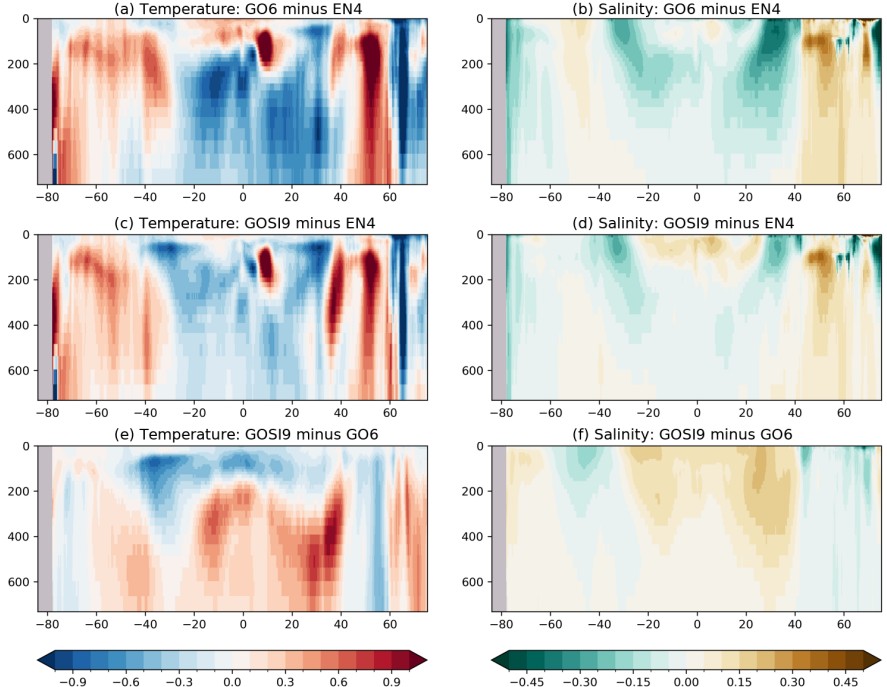

**Figure 5.** Zonal-mean potential temperature (K) and salinity : anomalies against 20-year climatology of EN4 v1.1 (Good et al., 2013) for the $\frac{1}{4}^{\circ}$ configuration for GOSI9 and GO6 and differences betweeen GOSI9 and GO6. Model fields are time-means over the third decade of integration.

## 4.2 Large scale biases

### 4.2.1 Zonal mean biases

For the next assessment we compute zonal and time mean quantities of temperature and salinity using the period 1996-2005, the last decade of the forced simulations. Figure 5 shows anomalies of these metrics computed for the $\frac{1}{4}^{\circ}$ configuration from both GO6 and GOSI9 from a reference 20-year climatology of EN4 v1.1, as well as anomalies between GOSI9 and GO6.

Both models exhibit a tendency to develop cold biases in the tropics and the Arctic, and warm biases in the extra-tropics and Southern Ocean. The cold bias between 200 m and 700 m present in GO6 between 30° S and 40° N is still present in GOSI9 but is significantly reduced, especially in the $\frac{1}{4}^{\circ}$ resolution (Fig. 5). The near-surface warm bias in the tropics is also greatly reduced, and there is a modest reduction of the cold bias in the Arctic. There is a slight degradation in the Southern Hemisphere extra-tropics, with a cold bias developing near 30° S, 50-100 m depth, and a slight increase in the warm bias between 40°-60° S below 200 m. Salinity biases reduce at most latitudes and depths, though a modest positive salinity bias develops in the shallow subsurface tropics. Overall, GOSI9 shows reduced temperature and salinity biases at most latitudes and depths.





We can attribute these changes to each of the development steps by comparing the simulations for each of the intermediate steps (Table 3). Figure 6 shows the incremental temperature and salinity anomalies that arise from updating the NEMO code from v3.6 to v4.0 (a, b), increasing the global mean chlorophyll concentration value to $0.1\,\mathrm{mg.m^{-3}}$ (c, d), and adjusting the
nn_htau parameter that controls TKE mixing penetration (e, f). Upgrading to NEMO 4.0 results in a warming between 100 and 700 m between 20° S and 40° N and in the Arctic (Fig. 6a), and is the change primarily responsible for reducing the cold bias observed in GO6 in these regions (Fig. 5a). It also acts to increase the salinity in the shallow tropical region (Fig. 6b), partially compensating the fresh biases in GO6 (Fig. 5b) and leading to the slight saline bias in GOSI9 (Fig. 5d). Increasing the chlorophyll concentration to $0.1\,\mathrm{mg.m^{-3}}$ has a cooling effect, which reduces subsurface warm biases in the tropics but slightly
increases cold biases in the extra tropics (Fig. 6c). The increase in chlorophyll does not affect the salinity. Adjusting the nn_htau parameter cools the subsurface between 50 m and 300 m and reduces the warm bias present in these layers between 30° and 40° S but in the forced GOSI9 configuration it introduces a cold bias centered at 100 m. This change also has a negligible effect on salinity. Its isolated effect is presented in Figure 6e,f. Despite the mixed impact on the forced GOSI9 configuration, the nn_htau adjustment was implemented with the purpose of keeping the same ocean configuration for forced and coupled
applications. In forced integration, GOSI9, the impact on the surface is more limited than in the coupled model as the SST is constrained by the bulk formulae but the impact on the subsurface is significant.

The more accurate tracer advection was found to result in reductions in the effective diffusivity, as defined by Megann (2018), of around 10%, as well as robust improvements in model biases, which were mainly attributed to increasing the order of the horizontal advection. Figure 7 shows the impact of the $4^{\mathrm{th}}$ order on the temperature in the top 700 m in the $\frac{1}{4}^{\circ}$ configuration.
Overall, using the $4^{\mathrm{th}}$ order tends to warm the ocean especially in the Atlantic basin. The impact is also seen on the western boundary currents (Gulf Stream and Kuroshio) where the steering is changed.

### 4.2.2 Spatial biases

Whilst the ability of a model to adequately represent globally or zonally integrated properties is important for applications such as climate projections and Earth system model studies, minimising regional biases is also critical, not only for those regions but
the biases can also affect climate and weather downstream. In this section we systematically assess the geographical distribution of biases.

For both GO6 and GOSI9, the three resolutions show a similar distribution of large-scale temperature and salinity biases. In GO6, temperatures are too warm at the surface and subsurface (Fig. 8 and Fig. 9) in the Southern Ocean, in the Tropics and in the Arctic Ocean. Changes made in GOSI9 significantly reduce the biases in the Arctic especially for the 1° and $\frac{1}{4}^{\circ}$
resolutions. This improvement is linked to an improved representation of Arctic sea ice in GOSI9 especially in Summer where the melting has been reduced and is in much better agreement with the observations. The warm SST bias in the Tropics present in GO6 is reduced in GOSI9, however the warm biases in the coastal upwellings in the East Pacific and East Atlantic are still present. At 100 m depth, GOSI9 is significantly cooler than GO6 (Fig. 9 and Fig. 5) as a result of the increased chlorophyll concentration. This change is net positive globally, reducing the warm bias centred at 100 m and present at all resolutions
in GO6 (Fig. 5). However it introduces regional cold biases, especially in the Indian Ocean. In the North Atlantic subpolar



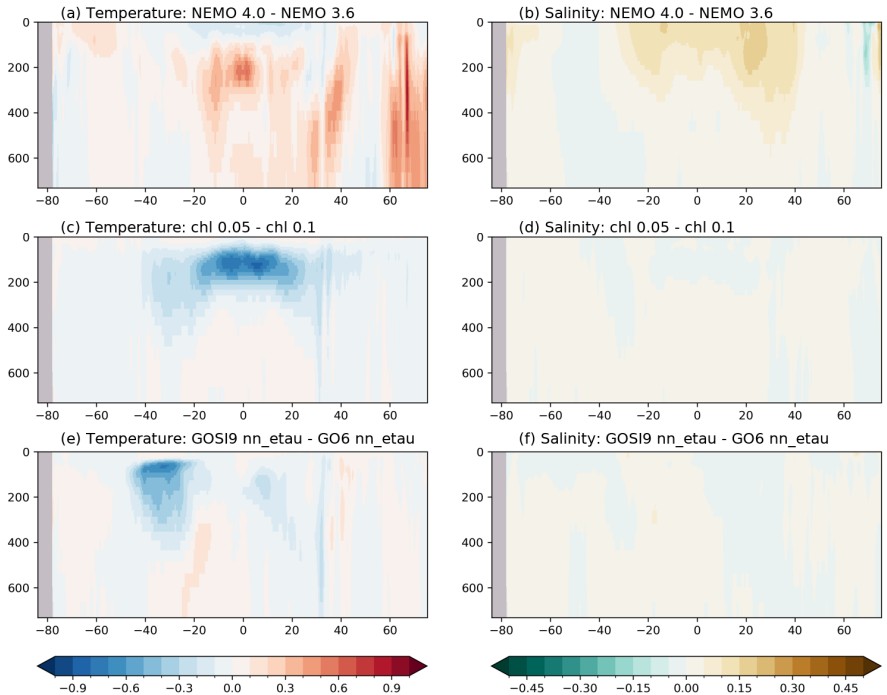

**Figure 6.** Zonal-mean potential temperature (K, left) and salinity (right) anomalies resulting from: updating the NEMO code from v3.6 to v4.0 (a, b); increasing the global mean chlorophyll concentration value to $0.1\,\mathrm{mg.m^{-3}}$ (c, d); adjustment of the nn_etau parameter that controls TKE mixing penetration (e, f). Model fields are time-means over the third decade of integration.

gyre, there is a dipole of cooling and warming, which is discussed in further details in section 5.2. Looking at the sea surface salinity, GO6 has large scale fresh biases between $40°$ S and $40°$ N (Fig. 10). Outside the Arctic, the largest biases occur in the locations where subtropical mode waters ventillate. These subtropical surface fresh biases extend to depths of $500\,\mathrm{m}$ in the $\frac{1}{4}°$ and $\frac{1}{12}°$ resolutions (Fig. 5). With the upgrade to NEMO4 and its more accurate bulk formulae, these fresh biases are

significantly reduced in GOSI9 and do not extend as deep (Fig. 5). In Arctic, the fresh biases present in GO6 are reduced in GOSI9. We note that comparison with EN4 in the Arctic is problematic due to the existence of artificial spatial patterns arising from the interpolation of sparse observational data. Figures 11 and 12 respectively show the annual minimum and maximum mixed layer depth biases compared to de Boyer Montégut et al. (2004). Summer minimum mixed layer depths are in good agreement with observations in all three models. There is an overall tendency towards a slight shallow bias of order 5-10 m,

with deep biases at the Equator and in a few regions around the Southern Ocean. Storkey et al. (2018) noted the too deep winter mixed layer in the North Atlantic subpolar gyre and in the Greenland-Iceland-Norway seas and in the Southern Ocean west of Drake Passage. These biases are still present in GOSI9 but with reduced amplitude. For the 1° resolution, biases in the NE Atlantic change from positive to negative, whilst the strong bias in the Labrador Sea is almost eliminated. Apart from these



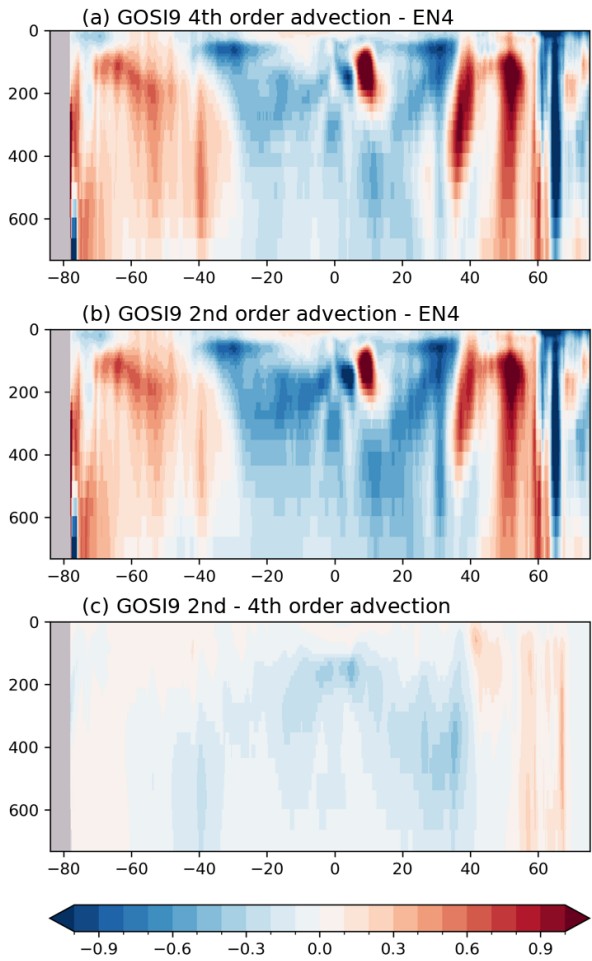

**Figure 7.** Potential temperature anomalies averaged over 0-700 m against EN4 v1.1 average 1995-2014 (Good et al., 2013) for GOSI9 and GOSI9 2nd order and difference between GOSI9 2nd order and GOSI9. Experiments GOSI9 and GOSI9 2nd order are detailed in Section 3 and Table 3. Model fields are time-means over the third decade of the integrations.

regions, differences in MLD between GO6 and GOSI9 can be observed in a latitudinal band around 40° S. This corresponds to the region where the TKE mixing depth has been reduced in GOSI9 (Section 2.5.8).

### 4.2.3 Sea ice

The seasonal cycle of sea ice area and volume is much improved in GOSI9 in both the northern and southern hemispheres (Fig. 13). This represents a marked improvement over GO6, for which the Arctic summer minimum was significantly lower than observed, especially for the $1°$ and $\frac{1}{4}^°$ resolutions. Ice representation is similar across the resolutions in GOSI9, so for conciseness we show only the assessment of $\frac{1}{4}^°$. Given the change of ice model from CICE5 to SI[3], it is difficult to attribute



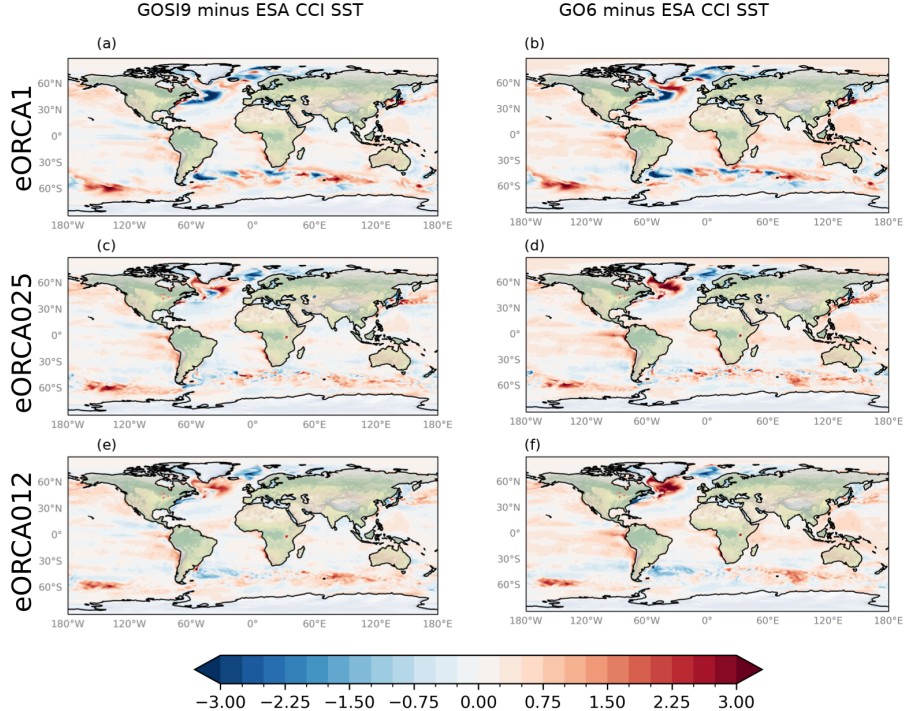

**Figure 8.** Sea surface temperature anomalies against 1996-2014 average ESA CCI (Merchant et al., 2014) for GOSI9 and differences between GOSI9 and GO6. Model fields are time-means over the third decade of the integrations.

these improvements more precisely. However, one significant change is the increase in albedo (section 2.5.2). Tested separately, it resulted in increased sea ice thickness and reduced melt in Summer Arctic sea ice (not shown). This is consistent with work from Rae et al. (2014) that shows how sensitive the sea ice models are to parameter changes that affect the surface radiation and highlights the Arctic sea ice as the most sensitive to snow albedo. Figure 14a-c shows the September mean for Arctic sea

ice concentration for the $\frac{1}{4}^{\circ}$ configuration at GOSI9, gridded observations from HadISST, and the $\frac{1}{4}^{\circ}$ configuration at GO6. The northern hemisphere distribution of sea ice in GOSI9 is in much better agreement with observations than GO6, though concentrations are still lower than observed over the central Arctic. Increasing the albedo in GOSI9 reduces the summer melt allowing a better representation of the minimum sea ice area. As a result of improving the sea ice area in Summer, the temperature and salinity biases are also reduced (Fig. 8 and Fig. 10). Sea ice distribution in the southern hemisphere is also

improved (Fig. 14d-f), though concentrations close to Antarctica in the Weddell, Bellingshausen, Amundsen and Somov Seas are all lower than observed. In the model low concentrations of ice not present in the observaations also extend east of the Weddell Sea into the Lazarev Sea.

    Overall, GOSI9 exhibits a more consistent and realistic representation of sea ice across the different resolutions than GO6+GSI8, for which the $\frac{1}{12}^{\circ}$ configuration was more realistic than the other two resolutions.



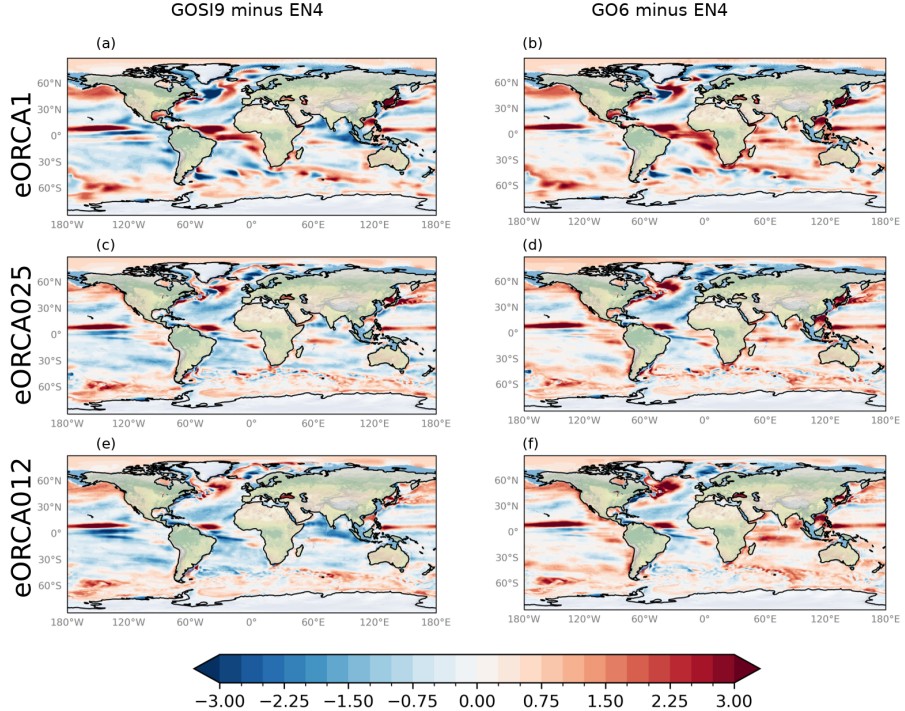

**Figure 9.** Potential temperature anomalies at 100 m against EN4 v1.1 average 1995-2014 (Good et al., 2013) for GOSI9 and differences between GOSI9 and GO6. Model fields are time-means over the third decade of the integrations.

## 5 Evaluation of model performance in specific regions of interest

Since GO6, concerns have been raised as to whether biases in specific regions of interest are limiting model performance, particularly in coupled forecasts and climate projections. In response JMMP have established process evaluation groups (PEGs) whose remit is to identify the causes of biases and to advise or deliver improvements to the model. To facilitate validations against observations and model comparison, key metrics have been identified and developed in a set of validation tools. These validation tools, MARINE_VAL, are available at https://github.com/JMMP-Group/MARINE_VAL. In this section we evaluate the improvements in model performance in three key regions.

### 5.1 Southern Ocean and Antarctic Circumpolar Current

As described in Section 2.5.6, significant work has been done to try to reduce large scale biases in the Southern Ocean in the coupled models with eddying ocean model resolution. In this section we show the impact of these changes in the forced model. The coupled results will be described in detail in a separate publication but here we also show results from a recent version of the HadGEM3 coupled model for illustration and comparison with the forced model results.



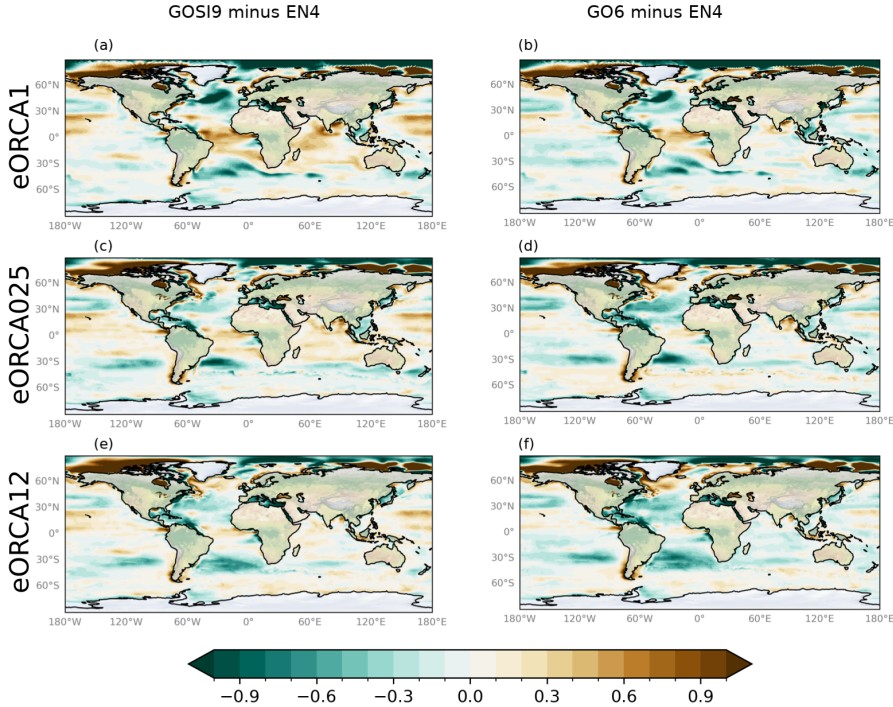

**Figure 10.** Sea surface salinity anomalies against EN4 v1.1 average 1995-2014 (Good et al., 2013) for GOSI9 and differences between GOSI9 and GO6. Model fields are time-means over the third decade of the integrations.

The biases are characterised by scalar metrics, detailed in the caption to Figure 15, in order to capture their time evolution. The immediate goal of the Southern Ocean package of changes in GOSI9 is to damp the overactive subpolar gyres in the Southern Ocean. The gyres in the Weddell Sea and Ross Sea are both too active at GO6 eddying resolutions and reduced in strength at GOSI9 (Figure 15 (b) and (c)). This change is primarily due to the Southern Ocean package, although the change to 4th order advection in the horizontal also has a beneficial impact (not shown). The results for the coupled model (eddy-permitting ocean resolution, present-day forcing) show that the biases are larger than for the forced models, due to coupled model feedbacks, and are significantly reduced by the Southern Ocean package of changes.

The metrics for the subpolar gyre strengths are based on the local maximum of the barotropic streamfunction. By construction, this is the sum of the transport in the circumpolar Antarctic Slope Current (ASC) and the recirculating transport in the gyre. A large part of the reduction in this transport metric is due to a reduction in the strength of the ASC. This has the additional effect of reducing an unrealistically large westward flow at the southern boundary of the Drake Passage and hence increasing the net eastward transport in the Drake Passage (Figure 15 (a)). The GO6 models at all resolutions have net Drake Passage transport which is too small compared to the estimate of Donohue et al. (2016), with the largest bias at eddy-permitting resolution. The biases are significantly reduced at GOSI9 for the eddying models. Again, the biases in the





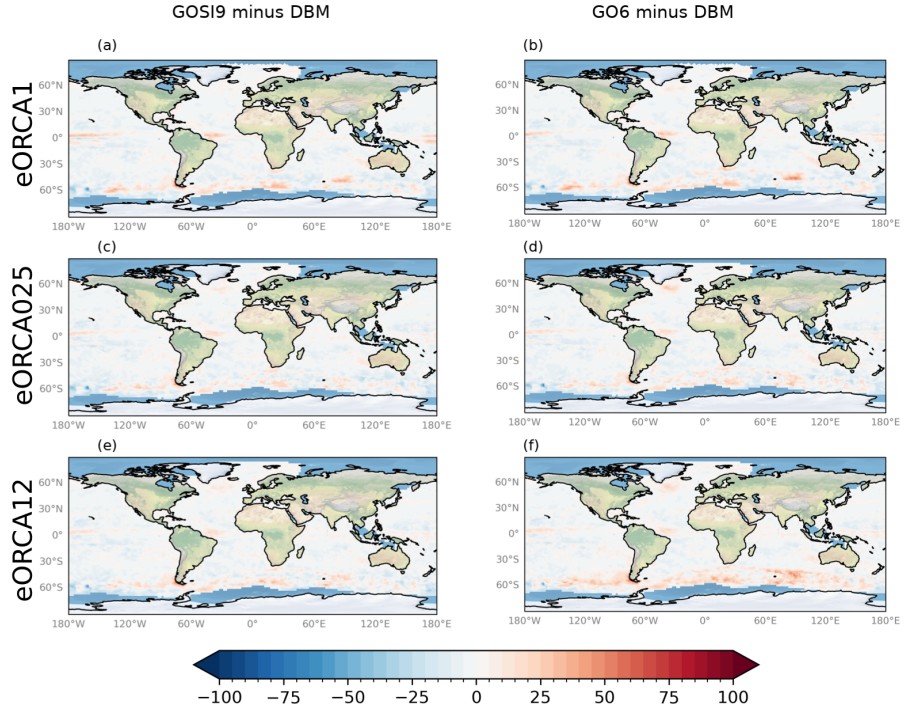

**Figure 11.** Annual minimum mixed layer depth (metres): anomalies against de Boyer Montégut et al. (2004) climatology for GOSI9 and GO6 and differences betweeen GOSI9 and GO6. Model fields are time-means over the third decade of integration.

coupled eddy-permitting model are larger than in the equivalent forced model, and there is a similar reduction with the use of the Southern Ocean package.

The very strong ASC in the eddying models tends to act as a barrier to exchange of water and ice between the Antarctic shelf and the open ocean (as also noted by Beadling et al. (2022)). This tends to lead to biases in the water mass properties
on the shelves. Two examples are shown here: the deep shelf water in the westernn Weddell sea tends to be too fresh in the coupled model integrations. This is a region of deep water formation and the freshening indicates that the process of deep water formation is tending to shut down in the model. The Southern Ocean package reduces this bias to some extent. The forced models have a smaller fresh bias in this region, which is somewhat reduced at GOSI9 compared to GO6. The other example is in the Amundsen Sea where relatively warm and salty circumpolar deep water (CDW) impinges onto the shelf. In
the coupled model at eddy permitting resolution, the strong ASC acts as a barrier to this and we see cold biases developing in this region, which are partially alleviated by the Southern Ocean package. In the forced models in this case we don't see the same biases: if anything the water in this region is slightly too warm, and there is little difference between GO6 and GOSI9.



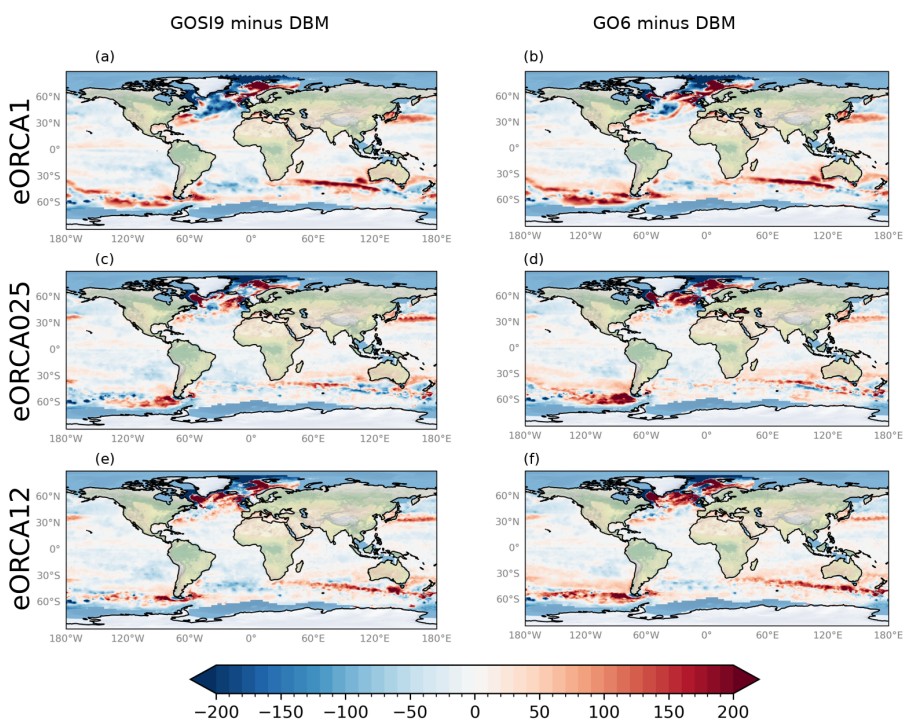

**Figure 12.** Annual maximum mixed layer depth (metres): anomalies against de Boyer Montégut et al. (2004) climatology for GOSI9 and GO6 and differences betweeen GOSI9 and GO6. Model fields are time-means over the third decade of integration.

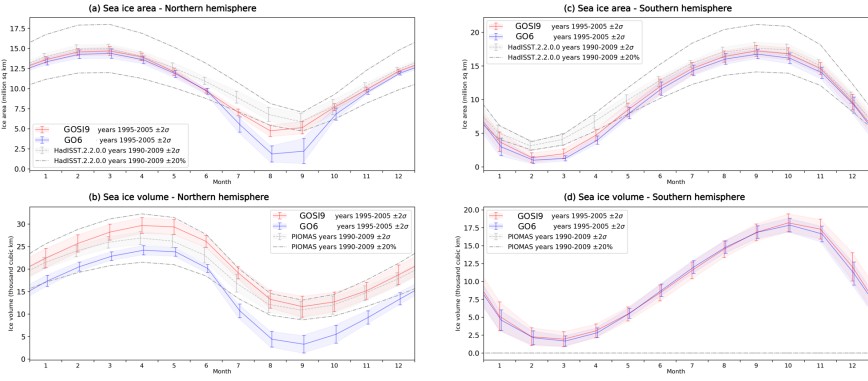

**Figure 13.** Mean seasonal cycles for integrated sea ice area for the Northern and Southern hemispheres for GOSI9 and GO6-GSI8.1. The meaning period is 1995-2014. Grey dashed lines show a climatology (mean and ±20%) of the HadISST analysis Titchner and Rayner (2014).



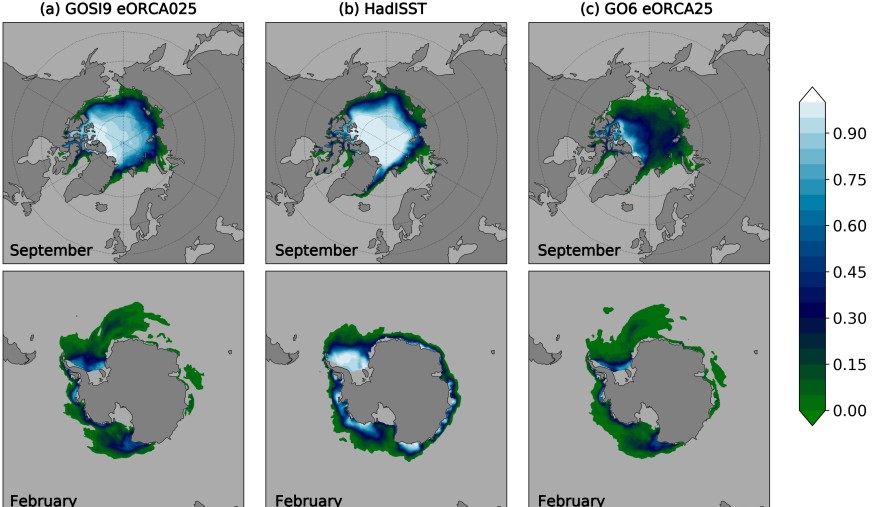

**Figure 14.** September mean for Arctic sea ice concentration (top) and February mean for Southern Ocean (bottom) averaged for 1995-2014 in GOSI9 (left), GO6-GSI8.1 (right) and HadISST (middle) analysis (Titchner and Rayner, 2014).

## 5.2 North Atlantic

The North Atlantic and its subpolar gyre is a key region for European weather and climate, and is also where some of the
largest biases on all three model resolutions occur. It is a dynamically active region where we see significant differences in
ocean currents and temperature and salinity biases between the different resolutions. In the GO6 $1°$ configuration a strong
cold bias in excess of -3 K manifests to the north of the Gulf Stream and east of Newfoundland (Fig. 8). All three resolutions
exhibit a warm bias throughout the rest of the subpolar gyre, which is in excess of 3 K in the eastern lobe of the subpolar gyre.
Changes introduced in GOSI9 have resulted in a substantial reduction of this warm bias in all three resolutions. The cold bias
at $1°$ remains similar in extent and magnitude, and where the warm bias has been corrected a more muted version of this cold
bias is also now present at $\frac{1}{4}°$. Strong SST gradients exist across the Gulf Stream separation and North Atlantic Current, so
biases of this magnitude can easily arise from biases in the position of the main current pathways in this region.

The Gulf Stream (GS) separation is known to be sensitive to resolution (Chassignet and Marshall, 2008; Marzocchi et al.,
2015). Chassignet and Marshall (2008) states that a resolution on the order of at least $1/10°$ is a necessary condition for
a western boundary current to realistically separate from the coast. Figure 16f shows GS separation latitude for the three
GOSI9 resolutions. The GS pathway is diagnosed using the $15°C$ isotherm at 200 m following (Seidov et al., 2019), and the
latitude of the GS at $72°W$ is displayed. In the $1°$ configuration, the GS separates too far north at $38.5°$ N and fails to deflect
northwards. The position of the North Atlantic Current (NAC) is calculated with a method similar to the one used to diagnose
the GS position but using the $10°C$ isotherm at 50 m depth. The latitude of the NAC at $41°W$ is shown in Fig. 16g. In the
$1°$ configuration the NAC position is $7°$ further south than observations. The poor representation of the GS and NAC in the
$1°$ model was already highlighted in GO6 (Storkey et al., 2018) and is a known issue in eddy-parameterising models (Zhang



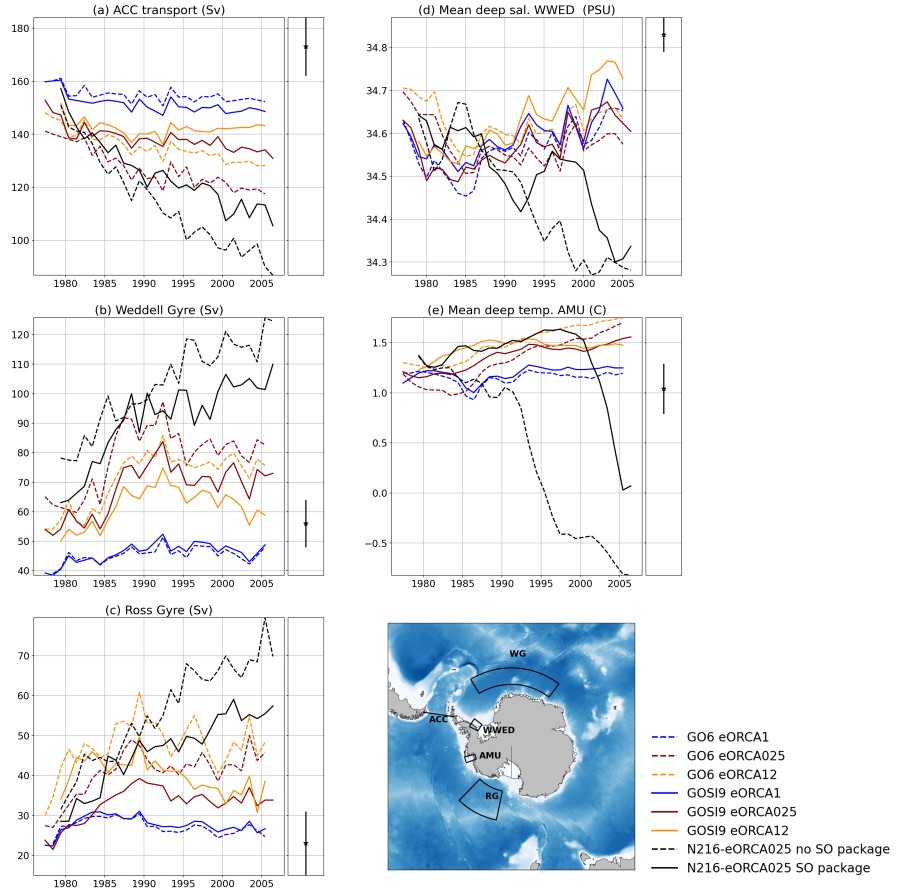

**Figure 15.** Time series of Southern Ocean metrics for the 30 years of GOSI9 and GO6 integrations compared with 30 years of a present-day forcing integration of two prototypes of the GC5 coupled model (N216-eORCA025) with and without the Southern Ocean Package described in Section 2.5.6 . The plotted quantities are annual means. Observational estimates and uncertainties are plotted as the black dots and lines to the right of the time series plots. From top left: a) the net eastward transport in the Drake Passage compared to the estimate of Donohue et al. (2016); b) the transport of the Weddell gyre as indicated by the maximum streamfunction in the WG box, compared to the estimate of Klatt et al. (2005); c) the transport of the Ross gyre as indicated by the maximum streamfunction in the RG box, compare to the estimate of Dotto et al. (2018); d) the salinity below 400 m spatially averaged over the WWED box in the western Weddell Sea; e) the temperature below 400 m averaged over the AMU box in the Amundsen Sea. For d and e, observational estimates are calculated from EN4.2.2.g10 profile dataset (Good et al., 2013) using means and standard deviations for all the profiles with data below 400 m in the defined boxes.

and Vallis, 2007). The lack of northward advection of warm and salty water from the GS results in fresh and cold biases off Newfoundland (Fig. 8,9,10a and 16c). In the $\frac{1}{4}^{\circ}$, as in the $1^{\circ}$, the GS separates too far north but the position of the NAC is noticeably improved compared with the $1^{\circ}$ (Fig. 16g). At $\frac{1}{12}^{\circ}$, the GS does not overshoot but separates further south than observations. The separation too far from the coast in the $\frac{1}{12}^{\circ}$ was already present in GO6 and in the coupled model based on




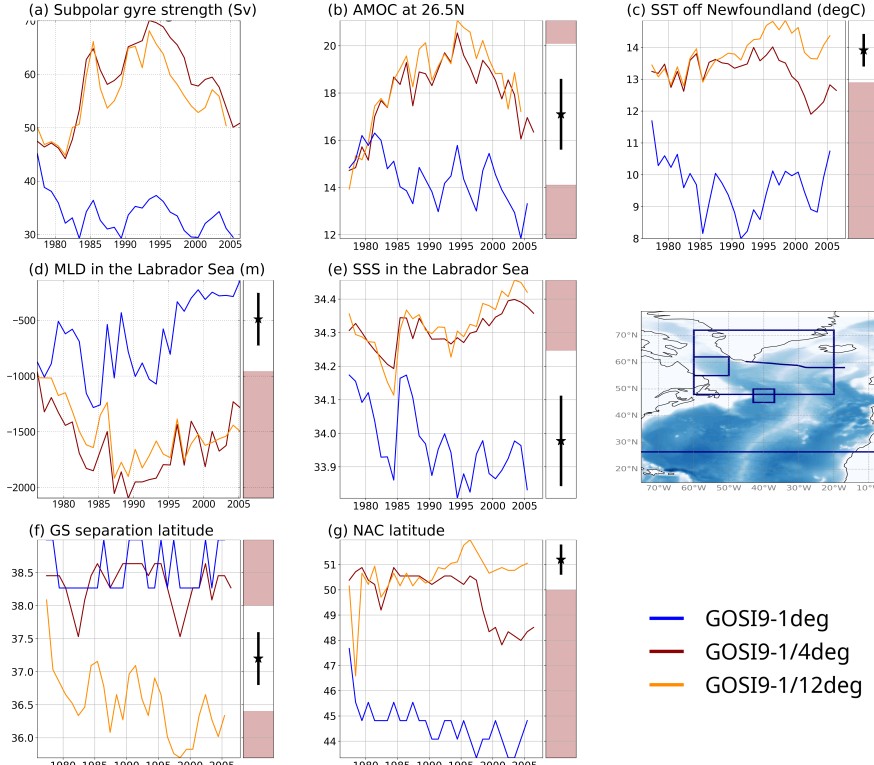

**Figure 16.** Time series of North Atlantic metrics for the 30 years of GOSI9 integrations. The plotted quantities are annual means. When available, observational estimates and uncertainties are plotted as the black dots and lines to the right of the time series plots. From top left: a) Subpolar gyre strength (Sv); b) AMOC at 26.5° N at maximum depth (Sv) compared to 2004-2008 RAPID array (Moat et al., 2020); c) Mean sea surface temperature over the box off Newfoundland (°C); d) Mean mixed layer depth in the Labrador Sea box (m) compared to de Boyer Montégut et al. (2004); e) Mean sea surface salinity in the Labrador Sea box ; f) Gulf Stream separation latitude at 72°W (° N) calculated using the 15°C isotherm at 200 m following Seidov et al. (2019). g) North Atlantic Current latitude at 41°W (° N) calculated using at 10°C isotherm at 50 m depth following similar method used for the Gulf Stream separation. For c,e,f and g, observational estimates are calculated from EN4.2.2.g10 profile dataset (Good et al., 2013) using means and standard deviation

GO6 (Grist et al., 2021). The pathway of the NAC in $\frac{1}{12}^{\circ}$ is better represented than in the lower resolution models and brings warmer and saltier water into the North Atlantic subpolar gyre (Fig. 8 and 10). Compared with GO6, there is no significant change in the GS separation latitude. However, GOSI9 exhibits a southward shift of the NAC compared with GO6. This is in better agreement with observations, and is mainly due to the impact of the $4^{th}$ order advection on the steering around Grand

Banks (not shown). The additional GM poleward of 50° and the upgrade to NEMO4 also contribute to the southward shift but to a lesser extent (not shown). As a result of this southward shift of the NAC, the northward advection of heat and salt is reduced in GOSI9. It reduces the warm and salty bias present in the subpolar gyre in GO6 and highlighted in Treguier et al. (2005) and Marzocchi et al. (2015). Marzocchi et al. (2015) linked the warm and salty bias to the strength of the cyclonic




subpolar gyre. However, the reduction of the temperature and salinity biases in GOSI9 is not associated with a change in the
strength of the SPG (not shown).

The time series of the annual-mean AMOC at 26°5 N are shown in Fig. 16b. Observations from the RAPID array (Moat
et al., 2020) between 2004-2018 estimate the transport to be 17.1 Sv with an estimated annual-mean rms uncertainty of 1.5
Sv. Transport estimates prior to RAPID suggest that the AMOC was slightly stronger during the 1980s and 1990s, with
net transport from a 1992 hydrographic section calculated as 19.4 Sv (Bryden et al., 2005). Annual mean AMOC transports
in the $\frac{1}{4}°$ and $\frac{1}{12}°$ GO6 models were unrealistic (Storkey et al., 2018), peaking at 26 Sv and 29 Sv respectively in the mid
1990s. Values in 2005 remained outside of observations, at 21 Sv and 23 Sv. In contrast, 1° showed excellent agreement with
observations, with a peak transport of around 20 Sv in the mid-late 1990s, reducing to 17 Sv in 2005. Improvements in GOSI9
have reduced the AMOC in $\frac{1}{4}°$ and $\frac{1}{12}°$ so that both now peak at 20-21 Sv in 1995 and reduce to values around 17 Sv in 2005.
The reduction in AMOC is associated with a decrease of the deep bias in the subpolar gyre and GIN seas mixed layer depth
(Fig. 12). However, the convective overturning in the Labrador Sea is still too high suggesting that the AMOC in $\frac{1}{4}°$ and $\frac{1}{12}°$
is driven too much by excessive deep mixing in the Labrador Sea as previously noted by Megann et al. (2014). Unfortunately,
AMOC volume transport in 1° is now too weak, around 16 Sv in 1995 and 13-14 Sv in 2005. Sensitivity experiments were
carried out with 1° to understand the changes in AMOC strength from GO6 to GOSI9, and the changes in AMOC are mainly
driven by the increased albedos in GOSI9 and the improved the sea ice cover (not shown). We speculate that in GO6 1° may
have been getting the right answer for the wrong reason, with the unrealistically low sea ice coverage in late summer-early
winter exposing the surface of the ocean and leading to strong heat loss, strong deep convection, and a more vigorous AMOC.
There are less than two years of overlap between the RAPID observation-based estimates of the strength of the AMOC and the
reference simulations. This will be addressed in the next development cycle, for which we will update the forcing product used.
It is also worth noting that the strength of the AMOC in numerical simulations is known to be sensitive both to the choice of
forcing product and to the strength of numerical mixing at tropical and subtropical latitudes (e.g. Megann and Storkey, 2021),
so care should be taken not to try to overfit transport in forced ocean models to the observations.

## 5.3 North West Pacific

The Kuroshio is the Western boundary current in the North Pacific subtropical gyre and plays an important role in transporting
heat poleward. Following methods described in Qiu et al. (2014) and Qiu and Chen (2005), the position and strength of the
Kuroshio is calculated for the 30-y GOSI9 integrations at all resolution and compared with sea surface height observations
from CMEMS for the period 1994-2006 (https://doi.org/10.48670/moi-00149). The position of the Kuroshio extension in the
1° configuration (38° N) and in the $\frac{1}{4}°$ configuration (37° N) is further north than in the observations (35° N) while the $\frac{1}{12}°$
configuration is in better agreement with the observations (Fig. 17a). An et al. (2023) shows that coupled models with eddy rich
ocean models better depict the Kuroshio extension dynamic and thermal structure than models with lower resolution ocean. Guo
et al. (2003) show that as the resolution increases, the path, intensity, and vertical structure of the Kuroshio improves. They
observed a northwards overshoot with the lower resolution model. They highlight that with increased resolution, the better
representation of topography results in better reproduction of the interaction between baroclinicity and bottom topography.



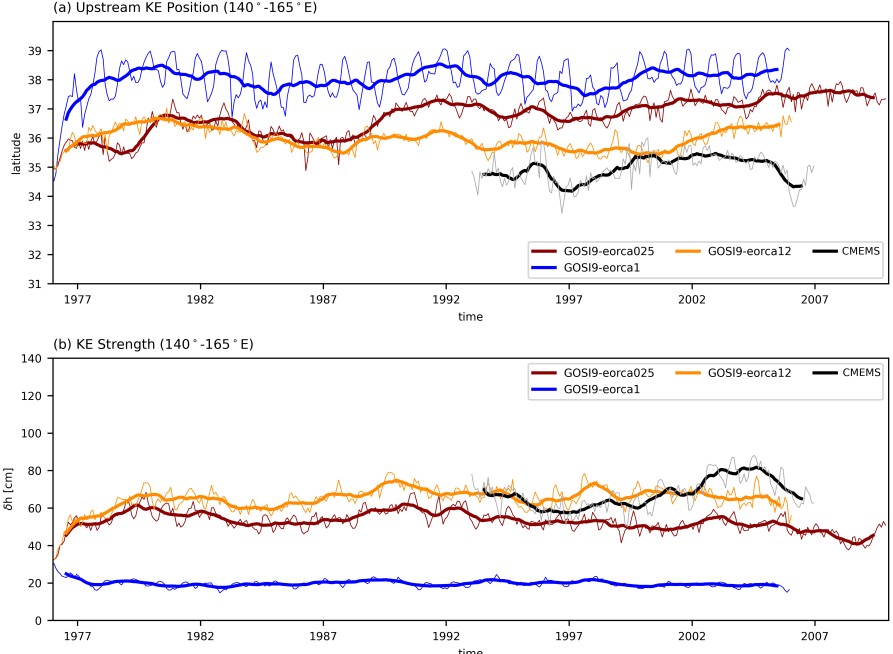

**Figure 17.** Time series of Kuroshio metrics for the 30 years of GOSI9 integrations and for 13 years of observations from Altimeter satellite gridded Sea Level Anomalies (E.U. Copernicus Marine Service Information (CMEMS). Marine Data Store (MDS). DOI: 10.48670/moi-00149). Annual (thick lines) and monthly (thin lines) are plotted. (a) Kuroshio extension zonal averaged latitude position in 140°-165°E calculated following Qiu et al. (2014); Qiu and Chen (2005). (b) Kuroshio extension strength calculated using the sea surface height difference across the Kuroshio extension as in Qiu et al. (2014).

In addition to a better representation of the position of the Kuroshio extension, in the $\frac{1}{12}^{\circ}$ configuration, the intensity of the current is in better agreement with the observations than in the 1° and $\frac{1}{4}^{\circ}$ configurations (Fig. 17b). It is noted than in the 1°, the intensity of the Kuroshio is very weak compared to the observations. The subarctic frontal zone formed by the convergence of cold water from the Oyashio and warm water from the Kuroshio is shifted northward in the two lower resolution models and this results in a warm anomaly (Fig. 9).

## 6 Summary and discussions

GOSI9 is the latest UK global ocean and sea ice configuration developed by the JMMP partnership (Met Office, National Oceanography Centre, British Antarctic Survey, and Centre for Polar Observation and Modelling), superseding GO6 (Storkey et al., 2018). GOSI9 is a traceable hierarchy of three horizontal resolutions 1°, $\frac{1}{4}^{\circ}$ and $\frac{1}{12}^{\circ}$, and is based on the NEMO version 4.0.4 code (Madec and system team, 2019). The upgrade to NEMO 4.0.4 includes a new sea ice model SI[3] (Sea Ice modelling Integrated Initiative) and faster integration achieved through the use of partially implicit schemes that allow a significant



increase in the length of the time step. Other developments include the upgrade to TEOS10 equation of state, reduction in
numerical mixing, and improved representation of the Southern Ocean. The interactive icebergs used in GO6 (Bigg et al.,
1997; Martin and Adcroft, 2010) are switched off in GOSI9 due to stability issues, especially while testing with the coupled
model. It is replaced by an iceberg melt climatology. The impact of these changes, principally in the $\frac{1}{4}^{\circ}$ are presented. The
upgrade to NEMO 4.0.4 has a large impact with a significant reduction of the temperature and salinity biases (Fig. 3b and Fig.
4b). Adopting the $4^{\text{th}}$ order horizontal and vertical advection reduces the numerical mixing, helping to minimise the cold bias
developing below 200 m (Fig. 3c and f). While the changes introduced to improve the representation of the Southern Ocean in
the $\frac{1}{4}^{\circ}$ and $\frac{1}{12}^{\circ}$ configurations have limited impact on the warm SST bias in the forced model, it results in a stronger and more
realistic ACC transport and a reduction of the temperature and salinity biases along the shelf of Antarctica (Fig. 15).

Tests with the coupled model were carried out in the early phase of GOSI9 development. It allowed us to integrate further
tuning required for the coupled model early during the development cycle. In particular, after testing with the coupled model,
change in TKE mixing depth and increased chlorophyll concentration were introduced to reduce subsurface biases. These two
changes have a positive impact on both forced and coupled models.

The results from the 30-year integrations forced by the CORE2 dataset are presented for the three resolutions and compared
against the GO6 integrations. In GOSI9, significant reductions in temperature and salinity drifts from initial condition are
realised. Below 200 m these are mainly due to the upgrade to NEMO 4.0.4 and $4^{\text{th}}$ order advection and in the top 200 m
primarily as a result of the change in chlorophyll 3. The global warm SST bias is reduced in the tropics and Arctic whilst
minimal changes are observed in the Southern Ocean. Large improvements in Arctic surface temperature and salinity are
linked to the improved sea ice representation. In particular, the excessive and unrealistic Arctic summer sea ice melt in GO6 is
significantly improved in GOSI9 14, and can be attributed to the change in the sea ice model to SI[3] and to the higher albedos
which increased sea ice thickness.

The next round of development will focus on improving model fidelity in the North Atlantic and will include work already
carried out to improve the representation of the Nordic overflows.Bruciaferri et al. (2024) combined the idea of Colombo
(2018) with the multi-envelope approach of Bruciaferri et al. (2018) to successfully implement localised terrain following
coordinates in the Nordic overflow region. Their generalised approach has been tested with the GOSI9 $\frac{1}{4}^{\circ}$ configuration and is
shown to significantly improve the realism of the Nordic overflows in simulations, reducing spurious cross-isopycnal mixing in
this region of strong gravity currents. However, large scale salinity biases along the bathymetry of the subpolar gyre impact the
mass properties of the water cascading. Improving the salinity bias and using appropriate coordinates are both key to improving
the representation of the Nordic overflows. Another key challenge is to realistically represent the separation latitude of the Gulf
Stream. In the GOSI9 configuration, increasing the resolution is shown to improve the realism of the Gulf Stream separation
latitude and the North Atlantic Current path. However, even with the $\frac{1}{12}^{\circ}$ it remains challenging. Work looking at the sensitivity
of the Gulf Stream to the vertical coordinate system is ongoing (Bruciaferri et al., 2022) with the same approach used for the
Nordic overflows and in parallel we plan to develop a $\frac{1}{12}^{\circ}$ configuration with a two-way nested AGRIF zoom with $\frac{1}{36}^{\circ}$ in the
North Atlantic.



In GOSI9, progress have been made in the Southern Ocean with increased ACC transport and reduced temperature and salinity biases on the Antarctic shelf but ice-shelf cavities have remained parameterised. For climate applications requiring open ice cavities, such as UK Earth System Model (UKESM) configurations (Smith et al., 2021), improving water mass properties on the shelf is essential. Using a 1° forced configuration, Hutchinson et al. (2023) show that explicitly simulating the circulation beneath the largest ice shelves improves the realism of the Antarctic continental shelf circulation. However, they note that the impact on the representation of the Antarctic Bottom Water is limited by the absence of realistic overflows. Recent work under the NERC funded ORCHESTRA project showed the merits of using a vertical sigma-coordinate around the Antarctic shelf region to greatly reduce temperature and salinity biases on the shelf and to preserve the density of Antarctic bottom water (Meijers et al., 2023). For the next configuration, we plan to open the ice-shelf cavities, taking advantage of the developments available at NEMO 4.2, and adopt the strategy used in the Nordic overflows using local terrain following coordinates to improve the Antarctic overflows.

In GOSI9, the bathymetries for each configuration originate from a different source (see 2.1). In this respect the hierarchy of resolutions is not fully traceable. For future GOSI configurations, work is ongoing to produce traceable bathymetry for each resolutions.

In GOSI9, the constant value used for the chlorophyll concentration was tuned to better match climatological values in the Tropics. For future configurations, the constant value will be replaced by a monthly chlorophyll concentration climatology derived from ocean colour observations that will better account for the variation in the solar penetration due to the large spatial and seasonal variability. For the coupled model, it will allow a consistent approach between the ocean models and the atmosphere models, where a varying chlorophyll concentration is already used to compute the albedo.

The vertical mixing closure schemes currently available in NEMO underestimate the impact of important sources of mixing such as Langmuir turbulence and maximum turbulence due to shear at the base of the mixed layer. These sources of mixing are important for the near surface mixing globally (Belcher et al., 2012). As part of the UK OSMOSIS project a more physically based mixing scheme has been developed and is being implemented in NEMO. We expect this will replace the existing TKE scheme in the next configuration.

*Code and data availability.* GOSI9 official release is available to download (Guiavarc'h and Storkey, 2024). For each configuration, it includes code, namelists, links to download the input files and scripts to run the configurations.

Each of the CORE2-forced 1976-2005 reference simulations have been archived to the CEDA archive (NERC's Environmental Data Service) and are available to download: eORCA1 (Blaker et al., 2023), eORCA025 (Guiavarc'h et al., 2023a), eORCA12 (Guiavarc'h et al., 2023b).

This study has been conducted using E.U. Copernicus Marine Service Information: Global Ocean Gridded L 4 Sea Surface Heights And Derived Variables Nrt (https://doi.org/10.48670/moi-00149).



*Author contributions.* CG prepared the manuscript with contributions from co-authors. CG, DS, and AB performed and analysed the main
assessment integrations. EB, AM, HH, MB, DCa, DCo, BS, PM, SM and BA were involved in the development of the GOSI9 configurations,
performed sensitivity experiments and assisted with the evaluation of the main integrations.

*Competing interests.* At least one of the (co-)authors is a member of the editorial board of Geoscientific Model Development.

*Acknowledgements.* This work was developed as part of the Joint Marine Modelling Programme (JMMP), a partnership between the Met
Office, National Oceanography Centre, British Antarctic Survey and Centre for Polar Observation and Modelling. The authors were supported
by the Met Office Hadley Centre Climate Programme funded by DSIT. This work was funded by the Met Office Weather and Climate Science
for Service Partnership (WCSSP) India project which is supported by the Department for Science, Innovation & Technology (DSIT) and by
the Met Office Climate Science for Service Partnership (CSSP) China project under the International Science Partnerships Fund (ISPF). The
contributions from the National Oceanography Centre were funded by the Natural Environment Research Council (NERC) under the Climate
Linked Atlantic Sector Science (CLASS) marine research programme (NE/R015953/1), as part of the Joint Marine Modelling Programme
(JMMP).



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
