# Peer review of "GOSI9: UK Global Ocean and Sea Ice configurations"

_EGUsphere, 2024_

## Author Comment (AC1)

**Response to reviewers of manuscript https://doi.org/10.5194/egusphere-2024-805**

Catherine Guiavarc'h[1], David Storkey[1], Adam T Blaker[2], Ed Blockley[1], Alex Megann[2], Helene Hewitt[1], Michael J Bell[1], Daley Calvert[1], Dan Copsey[1], Bablu Sinha[2], Sophia Moreton[1], Pierre Mathiot[3,1], and Bo An[4]

[1]Met Office, FitzRoy Road, Exeter EX1 3PB, UK
[2]Marine Systems Modelling, National Oceanography Centre, Southampton, SO14 3ZH, UK
[3]Univ. Grenoble Alpes/CNRS/IRD/G-INP, IGE, Grenoble, France
[4]State Key Laboratory of Numerical Modeling for Atmospheric Sciences and Geophysical Fluid Dynamics, Institute of Atmospheric Physics, Chinese Academy of Sciences, Beijing, 100029, China

**Correspondence:** Catherine Guiavarc'h (catherine.guiavarch@metoffice.gov.uk)

**1 Answer to anonymous referee #1**

We thank anonymous referee #1 for his comments. We have replied point by point below in blue.

**Line 28**: Not sure why Earth is not earth. We have corrected this in the revised manuscript

**Line 143**: The 2*10 does not make sense to me. Is the exponent missing? It should say $2\times10^{-3}$ m$^2$ s$^{-1}$. Corrected in revised manuscript

**Line 205**: It would be good to have more details on the machine used here since the speed of the cores will depend on which CPU's is installed. The integrations were done on a Cray XC40 supercomputer using intel Broadwell processors. We have added this information on the revised manuscript

**Line 268**: The figure reference is missing. I suspect that this is the figure in the supplement. The reference in the revised manuscript has been updated to point to the figure S1 in the supplement

**Line 440**: Westernn should be western. Corrected

**Figure 13**: This figure is very fuzzy when I print it and the difficult to read. The caption only mentions the sea ice area (top 2 plots) and not the sea ice volume (bottom 2 plots). There are also no references to the PIOMAS used for comparison of sea ice volume. We have reproduced figure 13 at a higher resolution we hope it solves the problem. We have updated the label to add the description of the sea ice volume and we have added a reference to PIOMAS (see Figure 13 below)

**Line 544**: I don't understand the 3 after chlorophyll. It refers to Figure 3, "Figure" was missing, updated in revised manuscript

**Line 544**: Likewise, the 14 after GOSI9. Likewise, "Figure" is missing, it refers to Figure 14, updated in revised manuscript

**2 Answer to Joakim Kjellsson**

We thank Joakim Kjellsson for his comments. We have replied point by point below in blue.

**Line 87**: The bathymetries have been derived in very different ways, using different sources and post processing methods. A motivation for this is needed. For example, why was the ORCA025 bathymetry smoothed but not the others? Why could the authors not derive the ORCA12 bathymetry and then coarse-grain it down to ORCA025 and ORCA1? We acknowledge this discrepancy, we have produced a new consistent set of bathymetries for the 3 resolutions for the next version of GOSI.

**Table 1**: I understand that one does not simply give NEMO a viscosity anymore but rather a velocity and length scale from which NEMO will compute a viscosity. But this Table makes it very hard to compare the values between configurations and to GO6 and GC3. I strongly recommend the authors to present the time step, lateral diffusivity, and lateral viscosity for the 1, 1/4, and 1/12 configurations in GO6 and GOSI9. This was done in Storkey et al. 2018. Since parameters vary horizontally, you could choose the value at the equator or some other reference latitude. We have updated the Table 1 with the value of the coefficients at the equator rather than the velocities. We only show the coefficient for GOSI9 as they are unchanged compared to GO6.

**Line 178**: The ocean will call SI3 each time step, meaning that the ice model time step is the same as the ocean model. Why was this choice made? One could also use the same ice-model time step for all configurations, which would likely make the higher resolutions (1/4, 1/12) faster, but perhaps that violates the CFL criterion in SI3? In NEMO the ice model timestep is the same as the SBC (surface boundary condition) timestep because SI3 runs from within the SBC. Whilst SI3 does not need to run every ocean timestep, the only way to stop this happening is to reduce the frequency of the SBC calls. We are not keen to reduce the temporal resolution of the fastest changing part of the system plus it would complicate the coupling with the atmosphere as SBC needs to be called on a coupling timestep.

**Line 205**: The purpose of the upgrade GO6 -> GOSI9 seems to have been two-fold: 1) Make a new model with smaller biases and 2) make a model that is faster. The authors discuss (1) a lot, but leave (2) out almost entirely. I would like to see a Table with the throughput and approximate cost for each model configuration, i.e. simulated years per day and core-hours per simulated year. If the numbers are also available for GO6, then we can judge if GOSI9 is faster as well as better, or if it's just better but not faster. We have added more informations in Table 2 and have expanded the section on model performances.

"The performances from GO6 configurations, as described in Storkey et al. (2018), and from GOSI9 configurations, as described in this paper, are summarized in Table 2. All the integrations were performed on a Cray X40 supercomputer using intel Broadwell processors. For the $\frac{1}{4}^{\circ}$ model eORCA025, the time step has been increased by 33% for the ocean-only (and by 50% for the coupled GC5 configuration). For the $\frac{1}{12}^{\circ}$ configuration eORCA12, the time step has been increased by 100%, allowing to produce 2 years of simulation per day on 6150 cores. For the $1^{\circ}$ configuration eORCA1, the time step has been increased by 33%. Note that for GO6, the "land suppression" option available in NEMO (Madec and system team, 2019) was only used for the $\frac{1}{12}^{\circ}$ configuration while it was used for the three resolutions in GOSI9. The "land suppression" option excludes much of the global land area from the calculations allowing to reduced the number of cores required. This explains the significant reduction of the number of cores used to perform the GOSI9 integrations for the $1^{\circ}$ and $\frac{1}{4}^{\circ}$ resolutions."

**Line 207**: One key to NEMO throughput can sometimes be the XIOS output server, which I am sure the authors have used, and I would guess that some effort went in to finding optimal settings to maximise throughout or minimise cost for 1/4 and 1/12 models. Some mention of this work would be of great interest to the community here. We did some work to optimise XIOS

settings for the configurations, testing different buffer size and number of XIOS servers to avoid have the outputs slowing down the models. We found that these settings did scale with the resolutions, so the requirement were dependant on the resolution but also on the level of outputs requested and on the version of XIOS used.

**Line 247**: "partial slip condition" is written twice corrected

**Line 268**: "Figure ??" refers to Figure S1. The reference in the revised manuscript has been updated to point to the figure S1 in the supplement

**Line 345**: There seems to be a big change in the temperature due to the upgrade to NEMO 4.0. I understand its near impossible to know exactly which part of the upgrade caused this, but some speculation would be welcome. Did the formulation of advection or diffusion change? Or is it mostly the upgrade from CICE to SI3? We can't indeed test separately every changes linked to the upgrade to NEMO 4.0. However, we can speculate that the large changes in temperature and salinity are likely linked to the improved bulk formulation available with AeroBULK package. The new computation of air density and reduction of approximation in the estimation of surface specific humidity of saturation would have impacted the evaporation especially.

**Line 390**: There seem to be a lot of changes in MLD but they are hard to see in Fig 12. I would recommend the authors to show zonal-mean MLD in the supplement. Perhaps that would reveal the shift around 40°S and the North Atlantic better? We have added a figure for the MLD zonal mean in the supplement, as suggested it highlights the differences around 40°S and around 60°N.

**Line 405**: The improvement in Antarctic sea ice (Fig 13) is very modest and within natural variability. Storkey et al. 2018 similarly showed very little change in ice extent but much more in ice volume. Is there a difference in ice volume between GO6 and GOSI9? Panels c and d in Figure 13 show the seasonal cycle of the sea ice volume compared to GO6. For the sea ice volume in Arctic, the increase is linked to the change in albedo however the impact on Antarctic sea ice volume is limited. The sea ice volume in the Southern Ocean in GOSI9 is similar to the volume in GO6.

**Fig 13**: This figure was hard to see on a printed A4. Suggest to make this larger when its time for publication! We have updated Figure 13 (see below), we hope it is improved.

**Fig 16d**: This figure is never referenced to, but I think it should be. ORCA1 produces a reasonable MLD while the higher resolutions overdo it. This is not uncommon in NEMO. Are 1/4 and 1/12 in GOSI9 better than in GO6, i.e. is the Labrador MLD less excessive? And what causes this bias? The convection becomes less excessive in 1/20° (Biastoch et al. 2021, doi: 10.5194/os-17-1177-2021, Fig 9 b,c) so perhaps its just a matter of having enough horizontal resolution to produce eddies in the Lab Sea? On the other hand, both 1/4 and 1/12 in GOSI9 include some scale-aware GM which one would hope reduces this problem a bit. The authors say that the GM had some impact on the NAC position. Have they looked into whether the inclusion of GM in 1/4 or 1/12 reduced the Lab Sea MLD? We agree with the comment, we have updated the text to add reference to Fig 16d and have added comment on the role of eddies not resolved even at 1/12 in the Labrador sea.

" The reduction in AMOC is associated with a decrease of the deep bias in the subpolar gyre and GIN seas mixed layer depth (Fig. 12). However, the convective overturning in the Labrador Sea is still too high in both the $\frac{1}{4}^{\circ}$ and $\frac{1}{12}^{\circ}$ configurations (Fig 16d). This suggests that the AMOC in $\frac{1}{4}^{\circ}$ and $\frac{1}{12}^{\circ}$ is driven too much by excessive deep mixing in the Labrador Sea as previously noted by Megann et al. (2014). The excessive deep convection in the Labrador Sea (Fig 16d) is common in ocean

numerical model (Courtois et al., 2017). Study by Chanut et al. (2008) highlights the crucial role played by eddies in the restratification process but even at $\frac{1}{12}^{\circ}$ these eddies are not fully resolved."

**Line 472**: "Compared to GO6" but we don't know what the position was for GO6. A number in the text or horizontal line in Fig 16 would help. We have added more details about the position of the NAC in GO6. "In GO6, the position of the NAC in $\frac{1}{4}^{\circ}$ and $\frac{1}{12}^{\circ}$ varies between $52°N$ and $56°N$ (not shown) further north than observations."

**Line 540 and Line 543**: Figure references are wrong. corrected

**Code availability**: The zenodo repository with namelists etc is zip file which I was unable to open on my Mac and a Windows machine. I suspect the file is broken. Thank you for pointing this out. We have created a new zenodo repository, it is now accessible (Guiavarc'h and Storkey, 2024)

**References**

Chanut, J., Barńier, B., Large, W., Debreu, L., Penduff, T., Molines, J. M., and Mathiot, P.: Mesoscale eddies in the Labrador Sea and their contribution to convection and restratification, Journal of Physical Oceanography, 38, https://doi.org/10.1175/2008JPO3485.1, 2008.

Courtois, P., Hu, X., Pennelly, C., Spence, P., and Myers, P. G.: Mixed layer depth calculation in deep convection regions in ocean numerical models, Ocean Modelling, 120, https://doi.org/10.1016/j.ocemod.2017.10.007, 2017.

Guiavarc'h, C. and Storkey, D.: JMMP-Group/GO_RELEASES: GOSI9 release, https://doi.org/10.5281/zenodo.13814369, 2024.

Madec, G. and system team, N.: Nemo Ocean Engine - version 4.0.1, Notes du Pôle de modélisation de l'Institut Pierre-Simon Laplace (IPSL): (27)., https://doi.org/10.5281/zenodo.3878122, 2019.

Megann, A., Storkey, D., Aksenov, Y., Alderson, S., Calvert, D., Graham, T., Hyder, P., Siddorn, J., and Sinha, B.: GO5.0: the joint NERC–Met Office NEMO global ocean model for use in coupled and forced applications, Geoscientific Model Development, 7, 1069–1092, https://doi.org/10.5194/gmd-7-1069-2014, 2014.

Storkey, D., Blaker, A. T., Mathiot, P., Megann, A., Aksenov, Y., Blockley, E. W., Calvert, D., Graham, T., Hewitt, H. T., Hyder, P., Kuhlbrodt, T., Rae, J. G., and Sinha, B.: UK Global Ocean GO6 and GO7: A traceable hierarchy of model resolutions, Geoscientific Model Development, 11, https://doi.org/10.5194/gmd-11-3187-2018, 2018.

Titchner, H. A. and Rayner, N. A.: The met office hadley centre sea ice and sea surface temperature data set, version 2: 1. sea ice concentrations, Journal of Geophysical Research, 119, https://doi.org/10.1002/2013JD020316, 2014.

Zhang, J. and Rothrock, D. A.: Modeling global sea ice with a thickness and enthalpy distribution model in generalized curvilinear coordinates, Monthly Weather Review, 131, https://doi.org/10.1175/1520-0493(2003)131<0845:MGSIWA>2.0.CO;2, 2003.

|  | eORCA1 | eORCA025 | eORCA12 |
|---|---|---|---|
| Lateral diffusion of momentum | laplacian | bilaplacian | bilaplacian |
| Lateral viscosity | 20,000 m$^2$ s$^{-1}$ | -1.5x10$^{11}$ m$^4$s$^{-1}$ | -1.25x10$^{11}$ m$^4$s$^{-1}$ |
| Isopycnal tracer diffusion | 1000m$^2$ s$^{-1}$ | 150m$^2$ s$^{-1}$ | 125m$^2$ s$^{-1}$ |

**Table 1.** Parameter changes between eORCA1, eORCA025 and eORCA12 configurations. The coefficient are calculated at the Equator, they reduce polewards, linearly with the grid size for the Laplacian and with the cube of the grid size for the biplacian. The coeficient are unchanged compared to GO6.

| Configuration | GO6 | | | GOSI9 | | |
|---|---|---|---|---|---|---|
| Resolution | 1 | 1/4 | 1/12 | 1 | 1/4 | 1/12 |
| time step (s) | 2700 | 1350 | 300 | 3600 | 1800 | 600 |
| number of core | 224 | 486 | 6237 | 156 | 344 | 6150 |
| number of simulated year per day | 18 | 2 | - | 24 | 2.8 | 2 |

**Table 2.** Summary of GO6 and GOSI9 performances for the 3 resolutions. The integrations were performed on a Cray XC40 supercomputer using intel Broadwell processors. The number of cores are for NEMO-SI3 only, it does not include XIOS cores. Note that we were not able to perform an additional integration to test GO6 eORCA12 performance in parallel to GOSI9 eORCA12.

[Figure]

**Figure 13.** Mean seasonal cycles for integrated sea ice area (a,c) and sea ice volume (b,d) for the Northern and Southern hemispheres for GOSI9 and GO6-GSI8.1. The meaning period is 1995-2014. In the panels representing the sea ice area (a and c), grey dashed lines show a climatology (mean and ±20%) of the HadISST analysis (Titchner and Rayner, 2014). Grey dashed lines in the Northern Hemisphere volume (b) plot show a climatology (mean and ±20%) of the PIOMAS reanalysis (Zhang and Rothrock, 2003).

[Figure]

**Figure S2.** Zonal mean mixed layer depth for the third decade of integration (1995-2005) for the GOSI9 and the GO6 configurations.

---

## Author Response (AR1)

**Response to reviewers of manuscript https://doi.org/10.5194/egusphere-2024-805**

Catherine Guiavarc'h[1], David Storkey[1], Adam T Blaker[2], Ed Blockley[1], Alex Megann[2], Helene Hewitt[1], Michael J Bell[1], Daley Calvert[1], Dan Copsey[1], Bablu Sinha[2], Sophia Moreton[1], Pierre Mathiot[3,1], and Bo An[4]

[1]Met Office, FitzRoy Road, Exeter EX1 3PB, UK
[2]Marine Systems Modelling, National Oceanography Centre, Southampton, SO14 3ZH, UK
[3]Univ. Grenoble Alpes/CNRS/IRD/G-INP, IGE, Grenoble, France
[4]State Key Laboratory of Numerical Modeling for Atmospheric Sciences and Geophysical Fluid Dynamics, Institute of Atmospheric Physics, Chinese Academy of Sciences, Beijing, 100029, China

**Correspondence:** Catherine Guiavarc'h (catherine.guiavarch@metoffice.gov.uk)

**1 Answer to anonymous referee #1**

We thank anonymous referee #1 for their comments. We have replied point by point to their comments and quoted the changes in the revised manuscript below.

**Line 28**: Not sure why Earth is not earth.

*Answer:* We have corrected this in the revised manuscript

**Line 143**: The 2*10 does not make sense to me. Is the exponent missing?

*Answer:* It should say $2 \times 10^{-3} \, \text{m}^2 \, \text{s}^{-1}$. Corrected in revised manuscript

**Line 205**: It would be good to have more details on the machine used here since the speed of the cores will depend on which CPU's is installed.

*Answer:* We have added this information on the revised manuscript.

*Change:* All the integrations were performed on a Cray X40 supercomputer using Intel Broadwell processors.

**Line 268**: The figure reference is missing. I suspect that this is the figure in the supplement.

*Answer:* The reference in the revised manuscript has been updated to point to the figure S1 in the supplement

**Line 440**: Westernn should be western.

*Answer:* Corrected

**Figure 13**: This figure is very fuzzy when I print it and the difficult to read. The caption only mentions the sea ice area

(top 2 plots) and not the sea ice volume (bottom 2 plots). There are also no references to the PIOMAS used for comparison of sea ice volume.

*Answer:* We have reproduced figure 13 at a higher resolution we hope it solves the problem. We have updated the label to add the description of the sea ice volume and we have added a reference to PIOMAS.

*Change:* see Figure 13 below

**Line 544**: I don't understand the 3 after chlorophyll.

*Answer:* It refers to Figure 3, "Figure" was missing, updated in revised manuscript

**Line 544**: Likewise, the 14 after GOSI9.

*Answer:* Likewise, "Figure" is missing, it refers to Figure 14, updated in revised manuscript

**2   Answer to Joakim Kjellsson**

We thank Joakim Kjellsson for his comments. We have replied point by point to the comment and quoted the changes in the revised manuscript below.

**Line 87**: The bathymetries have been derived in very different ways, using different sources and post processing methods. A motivation for this is needed. For example, why was the ORCA025 bathymetry smoothed but not the others? Why could the authors not derive the ORCA12 bathymetry and then coarse-grain it down to ORCA025 and ORCA1?

*Answer:* We acknowledge this discrepancy, we have produced a new consistent set of bathymetries for the 3 resolutions for the next version of GOSI.

**Table 1**: I understand that one does not simply give NEMO a viscosity anymore but rather a velocity and length scale from which NEMO will compute a viscosity. But this Table makes it very hard to compare the values between configurations and to GO6 and GC3. I strongly recommend the authors to present the time step, lateral diffusivity, and lateral viscosity for the 1, 1/4, and 1/12 configurations in GO6 and GOSI9. This was done in Storkey et al. 2018. Since parameters vary horizontally, you could choose the value at the equator or some other reference latitude.

*Answer:* We have updated the Table 1 with the value of the coefficients at the equator rather than the velocities. We only show the coefficient for GOSI9 as they are unchanged compared to GO6.

*Change:* see Table 1 below

**Line 178**: The ocean will call SI3 each time step, meaning that the ice model time step is the same as the ocean model. Why was this choice made? One could also use the same ice-model time step for all configurations, which would likely make the higher resolutions (1/4, 1/12) faster, but perhaps that violates the CFL criterion in SI3?

*Answer:* In NEMO the ice model timestep is the same as the SBC (surface boundary condition) timestep because SI3 runs from within the SBC. Whilst SI3 does not need to run every ocean timestep, the only way to stop this happening is to reduce the frequency of the SBC calls. We are not keen to reduce the temporal resolution of the fastest changing part of the system plus it would complicate the coupling with the atmosphere as SBC needs to be called on a coupling timestep.

**Line 205**: The purpose of the upgrade GO6 -> GOSI9 seems to have been two-fold: 1) Make a new model with smaller biases and 2) make a model that is faster. The authors discuss (1) a lot, but leave (2) out almost entirely. I would like to see a Table with the throughput and approximate cost for each model configuration, i.e. simulated years per day and core-hours per simulated year. If the numbers are also available for GO6, then we can judge if GOSI9 is faster as well as better, or if it's just better but not faster.

*Answer:* We have added more informations in Table 2 and have expanded the section on model performances.

*Change:* The performances from GO6 configurations, as described in Storkey et al. (2018), and from GOSI9 configurations, as described in this paper, are summarized in Table 2. All the integrations were performed on a Cray X40 supercomputer using Intel Broadwell processors. For the $\frac{1}{4}^{\circ}$ model eORCA025, the time step has been increased by 33% for the ocean-only configuration and by 50% for the coupled GC5 configuration. For the $\frac{1}{12}^{\circ}$ configuration eORCA12, the time step has been increased by 100%, allowing to produce 2 years of simulation per day on 6150 cores. For the $1^{\circ}$ configuration eORCA1, the time step has been increased by 33%. Note that for GO6, the "land suppression" option available in NEMO (Madec and system team, 2019) was only used for the $\frac{1}{12}^{\circ}$ configuration while it was used for the three resolutions in GOSI9. The "land suppression" option excludes much of the global land area from the calculations allowing to reduce the number of cores required. This explains the significant reduction of the number of cores used to perform the GOSI9 integrations for the $1^{\circ}$ and $\frac{1}{4}^{\circ}$ resolutions.

**Line 207**: One key to NEMO throughput can sometimes be the XIOS output server, which I am sure the authors have used, and I would guess that some effort went in to finding optimal settings to maximise throughout or minimise cost for 1/4 and 1/12 models. Some mention of this work would be of great interest to the community here.

*Answer:* We did some work to optimise XIOS settings for the configurations, testing different buffer size and number of XIOS servers to avoid have the outputs slowing down the models. We found that these settings did scale with the resolutions, so the requirement were dependant on the resolution but also on the level of outputs requested and on the version of XIOS used.

**Line 247**: "partial slip condition" is written twice

*Answer:* corrected

**Line 268**: "Figure ??" refers to Figure S1.

*Answer:* The reference in the revised manuscript has been updated to point to the figure S1 in the supplement

**Line 345**: There seems to be a big change in the temperature due to the upgrade to NEMO 4.0. I understand its near impossible to know exactly which part of the upgrade caused this, but some speculation would be welcome. Did the formulation of advection or diffusion change? Or is it mostly the upgrade from CICE to SI3?

*Answer:* We can't indeed test separately every changes linked to the upgrade to NEMO 4.0. However, we can speculate that the large changes in temperature and salinity are likely linked to the improved bulk formulation available with AeroBULK package. The new computation of air density and reduction of approximation in the estimation of surface specific humidity of saturation would have impacted the evaporation especially.

**Line 390**: There seem to be a lot of changes in MLD but they are hard to see in Fig 12. I would recommend the authors to show zonal-mean MLD in the supplement. Perhaps that would reveal the shift around 40°S and the North Atlantic better?

*Answer:* We have added a figure for the MLD zonal mean in the supplement, as suggested it highlights the differences around 40°S and around 60°N.

*Change:* Shallowing of MLD between GOSI9 and GO6, at around 40° S and at around 60° N, are more clearly visible in Figure S2, showing the mixed layer depth zonal mean for both models. See figure S2 below

**Line 405**: The improvement in Antarctic sea ice (Fig 13) is very modest and within natural variability. Storkey et al. 2018 similarly showed very little change in ice extent but much more in ice volume. Is there a difference in ice volume between GO6 and GOSI9?

*Answer:* Panels c and d in Figure 13 show the seasonal cycle of the sea ice volume compared to GO6. For the sea ice volume in Arctic, the increase is linked to the change in albedo however the impact on Antarctic sea ice volume is limited. The sea ice volume in the Southern Ocean in GOSI9 is similar to the volume in GO6.

**Fig 13**: This figure was hard to see on a printed A4. Suggest to make this larger when its time for publication!

*Answer:* We have updated Figure 13 (see below), we hope it is improved.

*Change:* See Figure 13 below

**Fig 16d**: This figure is never referenced to, but I think it should be. ORCA1 produces a reasonable MLD while the higher resolutions overdo it. This is not uncommon in NEMO. Are 1/4 and 1/12 in GOSI9 better than in GO6, i.e. is the Labrador MLD less excessive? And what causes this bias? The convection becomes less excessive in 1/20° (Biastoch et al. 2021, doi: 10.5194/os-17-1177-2021, Fig 9 b,c) so perhaps its just a matter of having enough horizontal resolution to produce eddies in the Lab Sea? On the other hand, both 1/4 and 1/12 in GOSI9 include some scale-aware GM which one would hope reduces this problem a bit. The authors say that the GM had some impact on the NAC position. Have they looked into whether the inclusion of GM in 1/4 or 1/12 reduced the Lab Sea MLD?

*Answer:* We agree with the comment, we have updated the text to add reference to Fig 16d and have added comment on the role of eddies not resolved even at 1/12 in the Labrador sea.

*Change:* " The reduction in AMOC is associated with a decrease of the deep bias in the subpolar gyre and GIN seas mixed layer depth (Fig. 12). However, the convective overturning in the Labrador Sea is still too high in both the $\frac{1}{4}^{\circ}$ and $\frac{1}{12}^{\circ}$ configurations (Fig 16d).This suggests that the AMOC in $\frac{1}{4}^{\circ}$ and $\frac{1}{12}^{\circ}$ is driven too much by excessive deep mixing in the Labrador Sea as previously noted by Megann et al. (2014). The excessive deep convection in the Labrador Sea (Fig 16d) is common in ocean numerical model (Courtois et al., 2017). Chanut et al. (2008) highlights the crucial role played by eddies in the restratification process but even at $\frac{1}{12}^{\circ}$ these eddies are not fully resolved."

**Line 472**: "Compared to GO6" but we don't know what the position was for GO6. A number in the text or horizontal line in Fig 16 would help.

*Answer:* We have added more details about the position of the NAC in GO6.

*Change:* In GO6, the position of the NAC in $\frac{1}{4}^{\circ}$ and $\frac{1}{12}^{\circ}$ varies between $52°$N and $56°$N (not shown) further north than observations.

**Line 540 and Line 543**: Figure references are wrong.

*Answer:* corrected in the revised manuscript

**Code availability**: The zenodo repository with namelists etc is zip file which I was unable to open on my Mac and a Windows machine. I suspect the file is broken.

*Answer:* Thank you for pointing this out. We have created a new zenodo repository, it is now accessible.

*Change:* Updated reference Guiavarc'h and Storkey (2024)

**References**

[revised manuscript text omitted]